# HORSESHOE SPLATTING: HANDLING STRUCTURAL SPARSITY FOR UNCERTAINTY-AWARE GAUSSIAN-SPLATTING RADIANCE FIELD RENDERING

**Feng Wu**[1]*,  **Tsai Hor Chan**[2]*,  **Yihang Chen**[1],  **Lingting Zhu**[1],
**Guosheng Yin**[1],  and  **Lequan Yu**[1]†
[1]University of Hong Kong    [2]University of Pennsylvania
{fengwu96, yihangc, ltzhu99}@connect.hku.hk
Tsaihor.Chan@PennMedicine.upenn.edu,{gyin, lqyu}@hku.hk

## ABSTRACT

We introduce Horseshoe Splatting, a Bayesian extension of 3D Gaussian Splatting (3DGS) that jointly addresses structured sparsity in per-splat covariances and delivers calibrated uncertainty. While neural radiance fields achieve high-fidelity view synthesis and 3DGS attains real-time rendering with explicit anisotropic Gaussians, existing pipelines do not explicitly encode structural sparsity in the covariance—e.g., axis-wise variances or pairwise correlations—leaving noise-dominated components insufficiently regularized. Uncertainty is likewise essential for trustworthy and robust novel-view prediction, yet most 3DGS variants remain deterministic. We place a global-local Horseshoe prior on the covariance scales, whose spike-at-zero and heavy-tails adaptively shrink irrelevant directions while preserving the salient structure. We fit the model with a factorized variational inference scheme that mirrors the Horseshoe's inverse-Gamma augmentation, enabling Monte Carlo rendering and pixel-wise posterior uncertainty with minimal overhead. Theoretically, we establish posterior contraction rates for the scale parameters and transfer them to the rendered image via a local Lipschitz mapping, providing guarantees that estimation error and predictive uncertainty diminish with data. Empirically, Horseshoe Splatting produces high-quality uncertainty maps while matching state-of-the-art 3DGS visual fidelity and runtime, yielding a practical, uncertainty-aware renderer that is robust to structured sparsity in the radiance field. The code is available at `https://github.com/HKU-MedAI/Horseshoe-Splatting`.

## 1 INTRODUCTION

Neural Radiance Fields (NeRFs) deliver high-fidelity novel view synthesis by optimizing a continuous volumetric scene function (Mildenhall et al., 2021), but their implicit networks are computationally intensive. The 3D Gaussian Splatting (3DGS) replaces per-ray network queries with explicit anisotropic Gaussians and a differentiable rasterization pipeline, enabling real-time, high-resolution rendering while retaining strong visual quality (Kerbl et al., 2023).

Despite its success, most 3DGS pipelines are deterministic, providing no notion of confidence, which is crucial under sparse views, occlusions, or out-of-distribution contents. Calibrated uncertainty in this setting can improve robustness by flagging unreliable regions and enabling active view selection or mapping via prioritizing uncertain areas. Furthermore, these methods also do not explicitly encode *structural sparsity* in the per-splat covariance (e.g., axis-wise variances or pairwise correlations), which can leave noise-dominated directions insufficiently regularized (Figure 1). Recent uncertainty-aware variants quantify aspects of scene ambiguity—e.g., semantic/posterior map variance (Wilson et al., 2025) or spatial depth uncertainty fields (Tan et al., 2025)—yet they do not target structured sparsity of the covariance itself, and hence cannot selectively suppress spurious variance or cross-axis

---

*Equal contribution.
†Corresponding author.

coupling in the Gaussian footprint. This gap limits both trustworthiness and robustness when signals are sparse or views are undersampled.

A principled way to address these issues is to bring Bayesian inference into 3DGS. While existing work has explored post-hoc pruning based on sensitivity (Hanson et al., 2025) or task-specific uncertainty modeling (Wilson et al., 2025; Tan et al., 2025), there remains little on hierarchical priors that *directly* regularize the per-splat covariance structure and yield coherent posterior uncertainty over rendered images.

To this end, we propose *structured* (global-local) priors on covariance scales. In particular, the Horseshoe prior provides high mass near zero (aggressive shrinkage of uninformative directions) and heavy tails (retention of salient structures) (Carvalho et al., 2009; Piironen and Vehtari, 2017). Placed on axis-wise scales—and when needed, on low-rank pairwise components—this hierarchy conforms to how 3DGS forms elliptical screen-space footprints, allowing the renderer to suppress noise along irrelevant axes while preserving sharp, data-supported anisotropy and enabling pixel-wise uncertainty via posterior sampling.

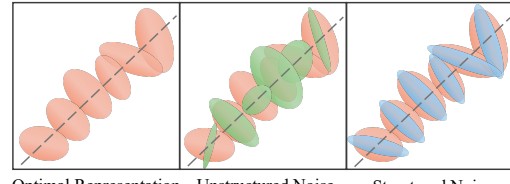

Optimal Representation   Unstructured Noise   Structured Noise

Figure 1: Unstructured and structured noise.

Building upon this, we introduce **Horseshoe Splatting**, a Bayesian 3DGS framework that imposes a Horseshoe prior on per-splat covariance scales and fits a factorized variational posterior. The variational family mirrors the Horseshoe's inverse-Gamma augmentation, enabling Monte Carlo rendering from the learned posterior and pixel-wise uncertainty maps at test time. To establish the theoretical soundness of our approach, we further prove that under a nonlinear observation model and a local Lipschitz renderer, the posterior over scales contracts at a near-minimax rate and transfers to image space, certifying that predictive uncertainty decreases with data.

Our contributions are: (1) A novel Bayesian formulation for 3DGS that imposes *structural sparsity* in the covariance via a global-local Horseshoe hierarchy; (2) A tractable, factorized variational inference scheme that supports Monte Carlo (MC)-based rendering and pixel-wise posterior uncertainty with minimal overhead; (3) Uncertainty estimates that improve reliability and downstream tasks (e.g., active view selection) without sacrificing speed; (4) Theoretically establishing posterior contraction for scales and rendered images under standard smoothness and identifiability conditions; and (5) State-of-the-art visual fidelity on standard benchmarks, while providing calibrated uncertainty.

## 2    RELATED WORKS

**Novel View Synthesis with NeRF and 3DGS.**    Novel view synthesis (NVS) has been significantly advanced by Neural Radiance Fields (NeRFs) (Mildenhall et al., 2021), which use continuous volumetric functions to achieve photorealistic rendering. Despite their success, NeRFs suffer from slow training and high computational cost (Müller et al., 2022). Consequently, numerous variants have emerged to improve rendering quality (Barron et al., 2021), accelerate training (Müller et al., 2022), or model unbounded scenes (Zhang et al., 2020a). More recently, 3D Gaussian Splatting (3DGS) (Kerbl et al., 2023) has become a highly efficient alternative, enabling real-time rendering and faster training by representing scenes with explicit Gaussian primitives. These Gaussians are typically initialized from a sparse Structure from Motion (SfM) point cloud (Schonberger and Frahm, 2016) and optimized with adaptive density control. Its superior performance has inspired rapid development, including extensions for dynamic scenes (Luiten et al., 2024; Sun et al., 2024; Kim et al., 2024), surface reconstruction (Guédon and Lepetit, 2024; Lyu et al., 2024), and applications in fields like autonomous driving (Bao et al., 2025).

**Uncertainty Estimation for Novel View Rendering.**    Quantifying prediction uncertainty is critical for trustworthy rendering in real-world applications (Amini-Naieni et al., 2024), yet it remains a challenge for most NeRF and 3DGS methods. In the context of NeRF, uncertainty has been explored via several avenues: Bayesian approaches (Shen et al., 2021; 2022), post-hoc methods (Goli et al., 2024), ensembles (Sünderhauf et al., 2023), and auxiliary networks (Xue et al., 2024; Amini-Naieni et al., 2024). However, these techniques often introduce significant computational overhead, complex training, or calibration difficulties. In contrast, uncertainty estimation for 3DGS is a less explored

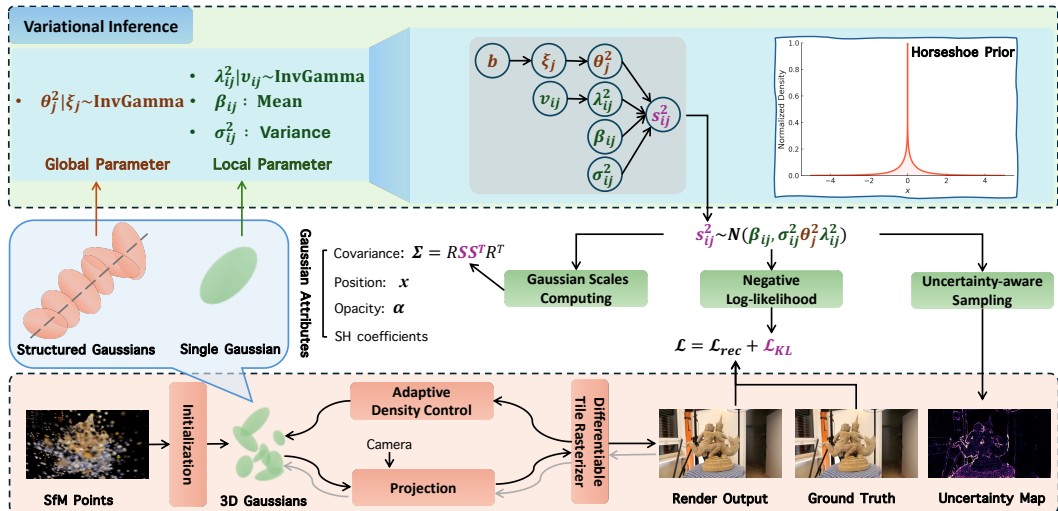

Figure 2: The framework of our proposed Horseshoe Splatting.

area. Recent works include FisherRF (Jiang et al., 2024), which uses Fisher information to estimate uncertainty, and several methods that apply variational inference (Savant et al., 2024; Li and Cheung, 2024) or uncertainty-aware regularization (Kim et al., 2024). Despite these advances, effectively modeling the anisotropic variance of each Gaussian primitive remains a key challenge, motivating our hierarchical Bayesian approach for more expressive uncertainty in 3DGS. Beyond radiance fields, non-RF 3D reconstruction routinely uses confidence or uncertainty (e.g., probability-volume variance or learned confidence in the multi-view stereo) to filter unreliable depth and guide fusion; our goals align with this tradition, but in the explicit 3DGS setting.

**Shrinkage Priors for Sparsification.** Shrinkage priors are Bayesian regularizers that pull noisy parameters toward zero while preserving strong signals. A prominent family of such priors is the global-local shrinkage class (Zhang et al., 2020b; Cadonna et al., 2020; Bhadra et al., 2017), which can be expressed as scale mixtures of Gaussians: each parameter $\theta_i$ is drawn from $\mathcal{N}(0, \tau^2 \lambda_i^2)$ with a global scale $\tau$ and individual local scales $\lambda_i$. This hierarchy enables adaptive sparsity: the global scale sets the overall shrinkage strength, while each local scale finely tunes how aggressively an individual coefficient is driven toward zero. Effective shrinkage priors exhibit two key properties: a sharp peak at zero to suppress noises, and heavy tails to avoid over-shrinking important signals. The Horseshoe prior (Carvalho et al., 2009) is a prime example, achieving these properties through a specific hierarchical construction. Despite their success in statistics and machine learning, their capability in tackling the sparsity issues in 3DGS remains underexplored.

## 3 METHODOLOGY

We present Horseshoe splatting, which imposes *structural sparsity* in the covariance via a global–local Horseshoe hierarchy to suppress the noise along irrelevant axes while preserving sharp, data-supported anisotropy. Figure 2 presents the overall framework with the detailed algorithm in Appendix E.

### 3.1 SPARSITY-INDUCED PRIOR ON 3D GAUSSIAN

**3D Gaussian Splatting.** In 3D Gaussian Splatting (Kerbl et al., 2023), a scene is modeled explicitly as a set of anisotropic Gaussians ("splats"), each parameterized by a center position $\mu_i \in \mathbb{R}^3$, an opacity $\alpha_i \in [0, 1]$, a color $c_i \in [0, 1]^3$ derived from Spherical Harmonics (SH), and a full covariance, $\Sigma_i = R_i S_i S_i^\top R_i^\top$, where $R_i \in SO(3)$ is a rotation matrix and $S_i = \text{diag}(s_{i1}, s_{i2}, s_{i3}) \succ 0$ is a diagonal scale matrix. During training, these parameters are optimized to minimize the reconstruction error of a set of posed input images, using a depth-aware rasterization that projects each 3D Gaussian to a 2D elliptical splat on the image plane. The rendered color at pixel $u$ is computed by front-to-back

compositing of overlapping splats,

$$\hat{I}(u) = \sum_{i=1}^{N} T_i\big(1 - \exp(-\alpha_i)\big)\, c_i, \quad T_i = \exp\Big(-\sum_{j<i} \alpha_j\Big).$$

This explicit, non-grid representation enables real-time, high-fidelity novel view synthesis without the discretization artifacts common in voxel or mesh-based methods.

**Horseshoe Prior on the Scaling Matrix.** For each splat $i$ and axis $j \in \{1, 2, 3\}$, we place a global–local Horseshoe prior on the diagonal scale $s_{ij}$ of $S_i = \mathrm{diag}(s_{i1}, s_{i2}, s_{i3})$. Let $\theta_j > 0$ be the (axis–wise) global shrinkage and $\lambda_{ij} > 0$ the local shrinkage. We model

$$s_{ij} \,|\, \lambda_{ij}, \theta_j \;\sim\; \mathcal{N}\big( \beta_{ij},\, \sigma_{ij}^2\, \theta_j^2 \lambda_{ij}^2 \big),$$

with learnable mean $\beta_{ij}$ and scale $\sigma_{ij} = \mathrm{softplus}(\rho_{ij})$. The half–Cauchy ($\mathrm{C}^+$) Horseshoe priors $\lambda_{ij} \sim \mathrm{C}^+(0, 1)$ and $\theta_j \sim \mathrm{C}^+(0, b)$ are expressed via the IG–IG (Inverse-Gamma) augmentation,

$$\lambda_{ij}^2 \,|\, \nu_{ij} \sim \mathrm{IG}\big(1/2,\, 1/\nu_{ij}\big),\; \nu_{ij} \sim \mathrm{IG}\big(1/2,\, 1\big),\; \theta_j^2 \,|\, \xi_j \sim \mathrm{IG}\big(1/2,\, 1/\xi_j\big),\; \xi_j \sim \mathrm{IG}\big(1/2,\, 1/b^2\big).$$

This yields the joint density

$$p(D, \boldsymbol{s}, \boldsymbol{\lambda}^2, \boldsymbol{\nu}, \boldsymbol{\theta}^2, \boldsymbol{\xi}) = p\Big(D \,|\, \{S_i\}_{i=1}^N\Big) \prod_{i=1}^{N} \prod_{j=1}^{3} f_{\mathcal{N}}\big(s_{ij} \,|\, \beta_{ij}, \sigma_{ij}^2 \theta_j^2 \lambda_{ij}^2\big) f_{\mathrm{IG}}\big(\lambda_{ij}^2 \,|\, 1/2,\, 1/\nu_{ij}\big)$$

$$\times\, f_{\mathrm{IG}}\big(\nu_{ij} \,|\, 1/2,\, 1\big) \prod_{j=1}^{3} f_{\mathrm{IG}}\big(\theta_j^2 \,|\, 1/2,\, 1/\xi_j\big) f_{\mathrm{IG}}\big(\xi_j \,|\, 1/2,\, 1/b^2\big),$$

with $\Sigma_i = R_i S_i R_i^\top$. This parameterization keeps the Gaussian prior on $s_{ij}$ (with variance modulated by $\lambda_{ij}^2 \theta_j^2$) while leveraging the Horseshoe's spike-and-heavy-tail behavior for adaptive shrinkage.

**Stochastic Variational Inference.** As exact Bayesian inference under the Horseshoe hierarchy is intractable, we adopt a mean–field variational family that mirrors the IG–IG augmentation. Writing the stacked variables as $\boldsymbol{s} = \{s_{ij}\}$, $\boldsymbol{\lambda}^2 = \{\lambda_{ij}^2\}$, $\boldsymbol{\nu} = \{\nu_{ij}\}$, $\boldsymbol{\theta}^2 = \{\theta_j^2\}_{j=1}^3$, $\boldsymbol{\xi} = \{\xi_j\}_{j=1}^3$, we use

$$q(\boldsymbol{s}, \boldsymbol{\lambda}^2, \boldsymbol{\nu}, \boldsymbol{\theta}^2, \boldsymbol{\xi}) = \prod_{i,j} q_N\big(s_{ij};\, \beta_{ij}, \sigma_{ij}^2\big) q_{\mathrm{IG}}\big(\lambda_{ij}^2;\, a_{ij}, b_{ij}\big) q_{\mathrm{IG}}\big(\nu_{ij};\, c_{ij}, d_{ij}\big)$$

$$\times \prod_{j=1}^{3} q_{\mathrm{IG}}\big(\theta_j^2;\, \alpha_j, \beta_j\big) q_{\mathrm{IG}}\big(\xi_j;\, \gamma_j, \delta_j\big),$$

with independent factors. The evidence lower bound (ELBO) is

$$\mathcal{L}(q) = \mathbb{E}_q\big[\log p(D \,|\, \{S_i\})\big] + \sum_{i,j} \mathbb{E}_q\Big[\log f_{\mathcal{N}}\big(s_{ij} \,|\, \beta_{ij},\, \sigma_{ij}^2 \theta_j^2 \lambda_{ij}^2\big)\Big]$$

$$+ \sum_{i,j} \mathbb{E}_q\Big[\log f_{\mathrm{IG}}\big(\lambda_{ij}^2 \,|\, 1/2,\, 1/\nu_{ij}\big) + \log f_{\mathrm{IG}}\big(\nu_{ij} \,|\, 1/2,\, 1\big)\Big]$$

$$+ \sum_{j=1}^{3} \mathbb{E}_q\Big[\log f_{\mathrm{IG}}\big(\theta_j^2 \,|\, 1/2,\, 1/\xi_j\big) + \log f_{\mathrm{IG}}\big(\xi_j \,|\, 1/2,\, 1/b^2\big)\Big] - \mathbb{E}_q\big[\log q(\boldsymbol{s}, \boldsymbol{\lambda}^2, \boldsymbol{\nu}, \boldsymbol{\theta}^2, \boldsymbol{\xi})\big].$$

The Gaussian prior term has a closed form,

$$\mathbb{E}_q\Big[\log f_{\mathcal{N}}\big(s_{ij} \,|\, \beta_{ij}, \sigma_{ij}^2 \theta_j^2 \lambda_{ij}^2\big)\Big] = -\tfrac{1}{2}\log(2\pi) - \tfrac{1}{2}\log\sigma_{ij}^2 - \tfrac{1}{2}\mathbb{E}_q\big(\log\theta_j^2\big) - \tfrac{1}{2}\mathbb{E}_q\big(\log\lambda_{ij}^2\big)$$

$$- \frac{\mathbb{E}_q\big[(s_{ij} - \beta_{ij})^2\big]}{2\,\sigma_{ij}^2}\, \mathbb{E}_q\Big(\frac{1}{\theta_j^2}\Big)\, \mathbb{E}_q\Big(\frac{1}{\lambda_{ij}^2}\Big),$$

where for $X \sim \mathrm{IG}(\alpha, \beta)$ (shape–rate), we use $\mathbb{E}(\log X) = \log\beta - \psi(\alpha)$ (where $\psi(\cdot)$ denotes the digamma function) and $\mathbb{E}(1/X) = \alpha/\beta$. and $\mathbb{E}(1/X) = \alpha/\beta$. We optimize $\mathcal{L}(q)$ by stochastic gradient ascent: $s_{ij}$ is sampled via the reparameterization $s_{ij} = \beta_{ij} + \sigma_{ij}\varepsilon$, $\varepsilon \sim \mathcal{N}(0, 1)$, while the IG factors admit low-variance pathwise gradients via implicit/transport reparameterizations for Gamma/IG families; the KL terms between IG factors remain analytic.

## 3.2 Integrating with Reconstruction Loss

We now combine our variational objective with the 3D Gaussian Splatting (3DGS) reconstruction loss to form the final training criterion. Let $D = \{I_u\}_{u=1}^{U}$ be the set of ground-truth pixel colors, and $\hat{I}_u(\{S_i\})$ the corresponding rendered colors (for a single sample of $\{S_i\}$). We define the reconstruction term as the expected negative log-likelihood under the variational posterior,

$$\mathcal{L}_{\text{rec}} \approx -\frac{1}{M} \sum_{m=1}^{M} \sum_{u=1}^{U} \ln f_{\mathcal{N}}\Big(I_u \mid \hat{I}_u^{(m)}, \sigma_u^2\Big).$$

where $\hat{I}_u^{(m)}$ is rendered with $\Sigma_i^{(m)}$ sampled from $q$, and $\sigma^2$ is a learnable variance parameter. We introduce a surrogate parameter $\rho$ such that $\sigma = \log(1 + e^\rho)$ to control the scale of prior variance.

The variational regularizer is the negative ELBO's KL component,

$$\mathcal{L}_{\text{KL}} = \text{KL}\big[q(\boldsymbol{s}, \boldsymbol{\lambda}^2, \boldsymbol{\theta}^2) \,\|\, p(\boldsymbol{s}, \boldsymbol{\lambda}^2, \boldsymbol{\theta}^2)\big].$$

Putting these together, the total loss minimized by stochastic gradient descent is

$$\mathcal{L}_{\text{total}} = \mathcal{L}_{\text{rec}} + \mathcal{L}_{\text{KL}} = -\mathbb{E}_q\big[\ln p(D \mid \{S_i\})\big] + \text{KL}\big[q(\boldsymbol{s}, \boldsymbol{\lambda}^2, \boldsymbol{\theta}^2) \,\|\, p(\boldsymbol{s}, \boldsymbol{\lambda}^2, \boldsymbol{\theta}^2)\big].$$

The prior term $p(\cdot)$ of the KL divergence acts as an automatic scaling factor that balances the regularization power of the Horseshoe assumption. In practice, we approximate both terms by Monte Carlo sampling from $q$ and use their analytic KL expressions under the inverse–Gamma factors. Minimizing $\mathcal{L}_{\text{total}}$ thus jointly drives accurate image reconstruction and enforces the Horseshoe-induced sparsity and uncertainty regularization on the Gaussian scales.

## 3.3 Posterior Inference and Uncertainty Estimation

After training, we generate posterior predictions by sampling *ancestrally* through the Horseshoe hierarchy, avoiding any auxiliary Gaussian variance parameters. For each splat $i$ and axis $j$, let the learned mean and base scale be $(\beta_{ij}, \sigma_{ij})$, and let the variational factors over the global–local shrinkage variables be $q_{\text{IG}}(\theta_j^2; \alpha_j, \beta_j)$ and $q_{\text{IG}}(\lambda_{ij}^2; a_{ij}, b_{ij})$ (with optional IG factors for the augmentation variables $\xi_j, \nu_{ij}$ when used). For $m = 1, \dots, M$, we draw $\theta_j^{2(m)} \sim q_{\text{IG}}(\theta_j^2), \lambda_{ij}^{2(m)} \sim q_{\text{IG}}(\lambda_{ij}^2), \varepsilon_{ij}^{(m)} \sim \mathcal{N}(0, 1)$, and set the scale sample $s_{ij}^{(m)} = \beta_{ij} + \sigma_{ij}\,\theta_j^{(m)}\lambda_{ij}^{(m)}\varepsilon_{ij}^{(m)}$, which is exactly the conditional Gaussian $p(s_{ij} \mid \theta_j^2, \lambda_{ij}^2)$ integrated against the variational posteriors of the Horseshoe scales (using the standard IG–IG augmentation of the half–Cauchy) (Makalic and Schmidt, 2015; Ghosal et al., 2000). We then form $\Sigma_i^{(m)} = R_i \,\text{diag}\big(s_{i1}^{(m)}, s_{i2}^{(m)}, s_{i3}^{(m)}\big) R_i^\top$, render each draw with the differentiable 3DGS rasterizer, and approximate the posterior predictive for each pixel $u$ as

$$p(\tilde{I}_u \mid D) \approx \frac{1}{M} \sum_{m=1}^{M} p\big(\tilde{I}_u \mid \{\Sigma_i^{(m)}, R_i, c_i, \alpha_i\}_{i=1}^{N}\big),$$

Per–pixel predictive mean, variance, and credible intervals are computed from the rendered samples $\{\tilde{I}^{(m)}\}_{m=1}^{M}$, yielding calibrated uncertainty maps while preserving the real–time rendering pipeline of 3DGS (Kerbl et al., 2023).

## 4 Theoretical Analysis

We develop the theoretical properties, in particular the posterior concentration rates of the scaling matrices and the resulting rendered scenes. Under scene sparsity, only a small subset of diagonal scale entries in the Gaussian covariances meaningfully contribute to rendering fidelity. Global–local shrinkage priors such as the Horseshoe concentrate posterior mass by aggressively shrinking noise-dominated coordinates toward zero while leaving large, signal-bearing coordinates relatively unshrunk. Consequently, the posterior over the scale vector contracts around the sparse ground truth at a near-minimax rate. Because the 3DGS renderer is a smooth map of these scales to images, this contraction propagates to the rendered scenes with (at most) a Lipschitz deformation of the rate.

We model the effective measurements used to infer the per–splat scales as a noisy nonlinear map $y = g(\boldsymbol{s}^*) + \varepsilon$, where $g$ summarizes how multi-view image evidence responds to changes in the per-splat scales (e.g., via differentiable rasterization statistics), and $\boldsymbol{y} \in \mathbb{R}^{p_y}$ denotes the observed data vector we use to infer the per-splat scale $\boldsymbol{s}$. We carry contraction from the scale space $\boldsymbol{s}$ to image space via a *local Lipschitz renderer* $\mathcal{R}$. The Horseshoe prior delivers adaptive sparsification and heavy–tailed protection, and its near–minimax contraction in sparse normal means extends to our nonlinear setting by local linearization and standard testing/prior–mass arguments.

**Assumption 1** (Nonlinear observation model). *Let $\boldsymbol{s}^* \in \mathbb{R}^{3N}$ be $k$-sparse. We observe $y = g(\boldsymbol{s}^*) + \varepsilon$, where $\varepsilon \sim \mathcal{N}\big(0, \frac{\sigma^2}{P} I_{p_y}\big)$ (where $I_{p_y}$ is the $p_y \times p_y$ identity matrix), $g : \mathbb{R}^{3N} \to \mathbb{R}^{p_y}$ is twice continuously differentiable on a neighborhood $\mathcal{N}(\boldsymbol{s}^*)$.*

**Assumption 2** (Local identifiability and curvature). *Let $J_* = \nabla g(\boldsymbol{s}^*)$. There exist $0 < \kappa_{\min} \leq \kappa_{\max} < \infty$ and $H_g < \infty$ such that for all $\boldsymbol{s} \in \mathcal{N}(\boldsymbol{s}^*)$,*

$$\kappa_{\min} \|\boldsymbol{s} - \boldsymbol{s}^*\|_2 \ \leq \ \|J_*(\boldsymbol{s} - \boldsymbol{s}^*)\|_2 \ \leq \ \kappa_{\max} \|\boldsymbol{s} - \boldsymbol{s}^*\|_2, \qquad \|\nabla^2 g(\boldsymbol{s})\|_{\mathrm{op}} \leq H_g,$$

*where $\|\cdot\|_{\mathrm{op}}$ denotes the operator norm. Consequently, $g$ admits the expansion $g(\boldsymbol{s}) = g(\boldsymbol{s}^*) + J_*(\boldsymbol{s} - \boldsymbol{s}^*) + r(\boldsymbol{s})$ with $\|r(\boldsymbol{s})\|_2 \leq \frac{H_g}{2}\|\boldsymbol{s} - \boldsymbol{s}^*\|_2^2$.*

**Assumption 3** (Horseshoe prior). *Each coordinate has the global–local Horseshoe hierarchy (half–Cauchy scales) in an inverse–Gamma mixture form:*

$$s_\ell \mid v_\ell \sim \mathcal{N}(0, v_\ell), \quad v_\ell \mid \lambda_\ell \sim \mathrm{IG}\Big(\tfrac{1}{2}, \tfrac{1}{\lambda_\ell}\Big), \quad \lambda_\ell \mid \theta^2 \sim \mathrm{IG}\Big(\tfrac{1}{2}, \tfrac{1}{\theta^2}\Big), \quad \theta^2 \sim \mathrm{IG}\Big(\tfrac{1}{2}, \tfrac{1}{\theta_0^2}\Big).$$

**Assumption 4** (Growth regime). *As $N, P \to \infty$, $k = o(3N)$ and $k \log\big(\frac{e\,3N}{k}\big) = o(P)$.*

**Assumption 5** (Local Lipschitz renderer). *Fix $R_i, c_i, \alpha_i$. The renderer $\mathcal{R} : \mathbb{R}^{3N} \to \mathbb{R}^M$, $\boldsymbol{I} = \mathcal{R}(\boldsymbol{s})$, is continuously differentiable $(C^1)$ on a neighborhood $\mathcal{N}(\boldsymbol{s}^*)$ and thus locally Lipschitz:*

$$\|\mathcal{R}(\boldsymbol{s}) - \mathcal{R}(\boldsymbol{s}')\|_2 \leq L_{\mathcal{R}} \|\boldsymbol{s} - \boldsymbol{s}'\|_2, \qquad \forall \boldsymbol{s}, \boldsymbol{s}' \in \mathcal{N}(\boldsymbol{s}^*),$$

*with $L_{\mathcal{R}} = \sup_{\boldsymbol{s} \in \mathcal{N}(\boldsymbol{s}^*)} \|\nabla \mathcal{R}(\boldsymbol{s})\|_{\mathrm{op}} < \infty$.*

**Lemma 1** (Horseshoe contraction for the linearized model). *Consider the surrogate linear model $y = g(\boldsymbol{s}^*) + J_*(\boldsymbol{s} - \boldsymbol{s}^*) + \tilde{\varepsilon}$, with $\tilde{\varepsilon} \sim \mathcal{N}(0, \frac{\sigma^2}{P} I_{p_y})$, under Assumptions 2 and 3. Then there exists $C > 0$ such that the posterior for $\boldsymbol{s}$ contracts at rate*

$$\varepsilon_{N,P}^2 \ = \ C\,\frac{\sigma^2}{P}\,\frac{k\,\log\big(\frac{e\,3N}{k}\big)}{\kappa_{\min}^2},$$

*and the posterior mean attains the same rate up to constants.*

Under Assumptions 1–2, we linearize $g$ at $\boldsymbol{s}^*$ and apply Lemma 1 to obtain Horseshoe posterior contraction for the scale vector after the variance rescaling $\sigma^2 \mapsto \sigma^2/P$ and with ambient dimension $d = 3N$; the constants $\kappa_{\min}$ and $H_g$ control, respectively, local identifiability and the Taylor remainder. The rendered–image rate then follows from Assumption 5 by a Lipschitz transfer, incurring at most a multiplicative factor $L_{\mathcal{R}}$. Theorems 1 and 2 formalize these two steps precisely.

**Theorem 1** (Contraction for sparse scales under nonlinear observations). *Under Assumptions 1–4, there exist constants $C, M > 0$ such that if the target radius satisfies $H_g\,\varepsilon_{N,P} = o(\kappa_{\min})$ (ensuring the second–order remainder is negligible at that scale), then*

$$\varepsilon_{N,P}^2 \ = \ C\,\frac{\sigma^2}{P}\,\frac{k\,\log\big(\frac{e\,3N}{k}\big)}{\kappa_{\min}^2}, \qquad \mathbb{E}_{\boldsymbol{s}^*}\Big[\Pi\big(\|\boldsymbol{s} - \boldsymbol{s}^*\|_2^2 > M\,\varepsilon_{N,P}^2 \mid y\big)\Big] \to 0.$$

*where $\Pi(\cdot \mid y)$ denotes the posterior distribution. Consequently, $\mathbb{E}_{\boldsymbol{s}^*}\|\mathbb{E}[\boldsymbol{s} \mid y] - \boldsymbol{s}^*\|_2^2 = O(\varepsilon_{N,P}^2)$.*

**Theorem 2** (Contraction for the rendered image). *Let $\boldsymbol{I} = \mathcal{R}(\boldsymbol{s})$ and $\boldsymbol{I}^* = \mathcal{R}(\boldsymbol{s}^*)$. Under Assumption 5, with $\tilde{\varepsilon}_{N,P} = L_{\mathcal{R}}\varepsilon_{N,P}$ from Theorem 1, there exists $M' > 0$ such that*

$$\mathbb{E}_{\boldsymbol{s}^*}\Big[\Pi\big(\|\boldsymbol{I} - \boldsymbol{I}^*\|_2 > M'\,\tilde{\varepsilon}_{N,P} \mid y\big)\Big] \to 0.$$

**Remark 2.1** (Adaptive sparsification of splats). *Theorem 1 shows Horseshoe shrinkage automatically drives near–zero scales to (near) zero at a near–minimax rate depending on $k$ and $\log(3N)$. This justifies pruning redundant splats and stabilizes optimization in sparse regions.*

**Remark 2.2** (Preservation of large structures). *Contraction together with the Horseshoe's heavy tails implies negligible bias for large coordinates; salient geometry (large $s_{ij}$) is preserved while noise is shrunk (van der Pas et al., 2014).*

**Remark 2.3** (Renderer sensitivity). *Image–space contraction holds with a Lipschitz loss of at most $L_{\mathcal{R}}$, which is finite under the differentiable rasterization/$\alpha$-compositing used by 3DGS on bounded neighborhoods.*

## 5 EXPERIMENTS

### 5.1 DATASETS AND EXPERIMENTAL SETTING

**Datasets.** Our experiments are conducted on two datasets: the Light Field **LF dataset** (Yücer et al., 2016) and **LLFF dataset** (Mildenhall et al., 2019). For the LF dataset which contains 8 scenes with dense 360° views, we adopt the setup from CF-NeRF (Shen et al., 2022) and evaluate on the `torch`, `basket`, `africa`, and `statue` scenes. For the LLFF dataset, we use all 8 forward-facing scenes, including `fern`, `flower`, `fortress`, `horns`, `leaves`, `orchids`, `room`, and `trex`, following established protocols (Shen et al., 2022; Li and Cheung, 2024).

**Task & Evaluation Metrics.** We focus on the **novel view synthesis (NVS)** task, where the objective is to render photorealistic images from new camera poses given only a sparse set of input views.

We assess model performance using standard metrics for both **image reconstruction quality** and **uncertainty estimation**. To evaluate the fidelity of synthesized views, we employ three metrics: Peak Signal-to-Noise Ratio (PSNR) for reconstruction accuracy, the Structural Similarity Index Measure (SSIM) for perceptual similarity, and the Learned Perceptual Image Patch Similarity (LPIPS) for perceptual distance. We evaluate the quality of our uncertainty predictions using two primary metrics. First, the Area Under the Sparsification Error (AUSE) curve, which measures the correlation between the estimated uncertainty and the true prediction error (MAE). Second, the Negative Log-Likelihood (NLL), which measures the likelihood of ground truth in the predictive distribution.

**Comparable Methods.** 1) **CF-NeRF** (Shen et al., 2022): A NeRF-based method that captures uncertainty by learning a conditional distribution over radiance values via a normalizing flow model. 2) **S-NeRF** (Shen et al., 2021): A Bayesian variant of NeRF that performs variational inference on the model's weights to derive uncertainty for both rendered color and depth. 3) **Bayes' Ray** (Goli et al., 2024): Introduces a spatial uncertainty field that perturbs input ray coordinates and propagates this geometric uncertainty through the radiance field using Laplace approximation. 4) **FisherRF** (Jiang et al., 2024): A 3DGS method that uses Fisher information to estimate uncertainty; 5) **Variational 3DGS** (Li and Cheung, 2024): A variational inference framework for 3DGS that introduces a multi-scale representation to explicitly model uncertainty. 6) **Ensemble GS**: Following Li and Cheung (2024), 10 3DGS models are trained with different subsets of initialization points from Structure from Motion (SfM) (Schonberger and Frahm, 2016) and different random seeds. The variance of the predictions between all models is regarded as the predictive uncertainty.

**Implementation Details.** Our method extends the official 3DGS PyTorch framework (Kerbl et al., 2023). We use standard SfM initialization and the original training schedules for all base 3DGS parameters. The learning rates for our variational local and global scale parameters are set to $1 \times 10^{-4}$ and $1 \times 10^{-5}$, respectively. At inference, uncertainty is estimated via 10 Monte Carlo samples. All experiments were run on a single NVIDIA RTX 3090 GPU.

### 5.2 EXPERIMENTAL RESULTS

**Quantitative Results.** We evaluate our method's performance in both uncertainty estimation and novel view synthesis, with results detailed in Table 1 and Table 2.

*Uncertainty Estimation.* For depth uncertainty on the LF dataset, our method achieves a state-of-the-art average AUSE of 0.18, outperforming all baselines. This is highlighted in the `basket` scene, where our 0.10 AUSE marks a 23% improvement over the next best method. For RGB image uncertainty, our method also excels. On the LF dataset, it delivers the best performance with a leading AUSE of 0.25 and a significantly lower NLL of −0.74. On the more challenging LLFF dataset,

Table 1: Depth uncertainty estimation (AUSE-MAE) performance on the LF dataset. The best result is in **boldface**, and the second-best is underlined.

| LF Dataset | africa | basket | statue | torch | **Average** |
|---|---|---|---|---|---|
| CF-NeRF (Shen et al., 2022) | 0.35 | 0.31 | 0.46 | 0.97 | 0.52 |
| S-NeRF (Shen et al., 2021) | 0.66 | 0.38 | 0.67 | 0.74 | 0.61 |
| Bayes' Ray (Goli et al., 2024) | 0.27 | 0.28 | **0.17** | **0.22** | 0.23 |
| FisherRF (Jiang et al., 2024) | 0.21 | 0.17 | 0.25 | 0.24 | 0.22 |
| Variational 3DGS (Li and Cheung, 2024) | 0.19 | 0.13 | 0.21 | 0.23 | 0.19 |
| Ensemble GS ($\times10$) | **0.16** | 0.22 | **0.17** | 0.26 | 0.20 |
| **Horseshoe Splatting (Ours)** | 0.19 | **0.10** | 0.20 | 0.24 | **0.18** |

Table 2: NVS and uncertainty estimation results on the LF and LLFF datasets. The best result is in **boldface**, and the second-best is underlined.

| Dataset | Method | Synthesized View Quality | | | Uncertainty Quality | |
|---|---|---|---|---|---|---|
| | | PSNR ↑ | SSIM ↑ | LPIPS ↓ | AUSE ↓ | NLL ↓ |
| LF | CF-NeRF (Shen et al., 2022) | 24.32 | 0.835 | 0.202 | 0.49 | -0.37 |
| | S-NeRF (Shen et al., 2021) | 20.21 | 0.761 | 0.248 | 0.62 | 1.32 |
| | FisherRF (Jiang et al., 2024) | 29.13 | 0.927 | 0.076 | 0.54 | 7.02 |
| | Variational 3DGS (Li and Cheung, 2024) | 27.39 | 0.914 | 0.101 | 0.26 | -0.30 |
| | Ensemble GS ($\times10$) | 27.64 | 0.902 | 0.088 | 0.29 | -0.34 |
| | **Horseshoe Splatting (Ours)** | **30.05** | **0.947** | **0.064** | **0.25** | **-0.74** |
| LLFF | CF-NeRF (Shen et al., 2022) | 21.74 | 0.782 | 0.190 | 0.48 | 0.58 |
| | S-NeRF (Shen et al., 2021) | 20.10 | 0.744 | 0.221 | 0.59 | 0.91 |
| | FisherRF (Jiang et al., 2024) | 25.34 | 0.849 | 0.125 | 0.51 | 7.05 |
| | Variational 3DGS (Li and Cheung, 2024) | 23.97 | 0.806 | 0.172 | 0.32 | 0.23 |
| | Ensemble GS ($\times10$) | 24.54 | 0.810 | 0.157 | **0.30** | 0.26 |
| | **Horseshoe Splatting (Ours)** | **25.86** | **0.864** | **0.110** | 0.31 | **0.14** |

it achieves the best NLL of 0.14, indicating a more accurate predictive distribution. These results confirm our model's ability to produce well-calibrated uncertainty across modalities and datasets.

*Novel View Synthesis.* As shown in Table 2, our method also outperforms others for rendering quality. On the LF dataset, it achieves a PSNR of 30.05, surpassing the closest competitor by a significant margin of over 0.9, while also leading in SSIM and LPIPS. On the more challenging forward-facing LLFF dataset, our method continues to excel, leading in all image fidelity metrics. This demonstrates that our approach not only provides superior uncertainty but also produces state-of-the-art, high-fidelity renderings.

**Qualitative Results.** Figure 3 visualizes rendering results on the `fern` scene. Our method achieves the best reconstruction quality, correctly rendering fine details like complex leaf structures. The error map shows that our errors are minimal and confined to inherently challenging areas, such as thin leaf edges and dark boundary lines. The main advantage of our method is shown in the uncertainty maps. Our uncertainty estimate exhibits the strongest correlation with the actual error map, especially in difficult regions like the plant's stem. In contrast, while FisherRF achieves good reconstruction, its uncertainty estimation is poor. This is because its Hessian-based approximation relies on depth information, which does not translate well to the RGB space, resulting in noisy and uninformative uncertainty. In summary, the visualization confirms that our method not only delivers superior reconstruction quality but also provides more accurate and reliable uncertainty estimates.

**Active View Selection.** A key application of reliable uncertainty is guiding data acquisition in active learning. We conduct an experiment on the LLFF dataset, starting with 10% of views and iteratively adding one view every 500 training steps based on uncertainty, up to 30% of total views. Specifically, we select the view with the highest integrated pixel-wise variance. We strictly enforce this budget alignment for all baselines, and the model is further trained for 7K steps after all images are chosen.

As shown in Table 4, our method significantly outperforms all baselines. This result confirms that our well-calibrated uncertainty is highly effective in identifying the most informative views, leading to greater data efficiency in reconstruction.

**Inference Time Analysis.** We report the inference speed of 3DGS-based uncertainty methods in Table 5. The inference time is measured as the average time to render 1000 frames on the `torch` scene of the LF dataset. At just 0.03 seconds per view, our method is by far the fastest, running

Table 3: Horseshoe Prior compared with Laplace Prior and Gaussian Prior.

| Dataset | Method | Synthesized View Quality | | | Uncertainty Quality | | |
|---|---|---|---|---|---|---|---|
| | | PSNR ↑ | SSIM ↑ | LPIPS ↓ | AUSE ↓ | NLL ↓ | Depth AUSE ↓ |
| LF | Laplace Prior | 30.04 | 0.942 | 0.065 | 0.37 | 10.58 | 0.19 |
| | Gaussian Prior | 30.01 | 0.941 | 0.067 | 0.38 | 9.15 | 0.18 |
| | Horseshoe Prior | **30.05** | **0.947** | **0.064** | **0.25** | **-0.74** | 0.18 |
| LLFF | Laplace Prior | 25.74 | 0.860 | 0.112 | 0.42 | 8.22 | – |
| | Gaussian Prior | 25.61 | 0.859 | 0.116 | 0.42 | 6.98 | – |
| | Horseshoe Prior | **25.86** | **0.864** | **0.110** | **0.31** | **0.14** | – |

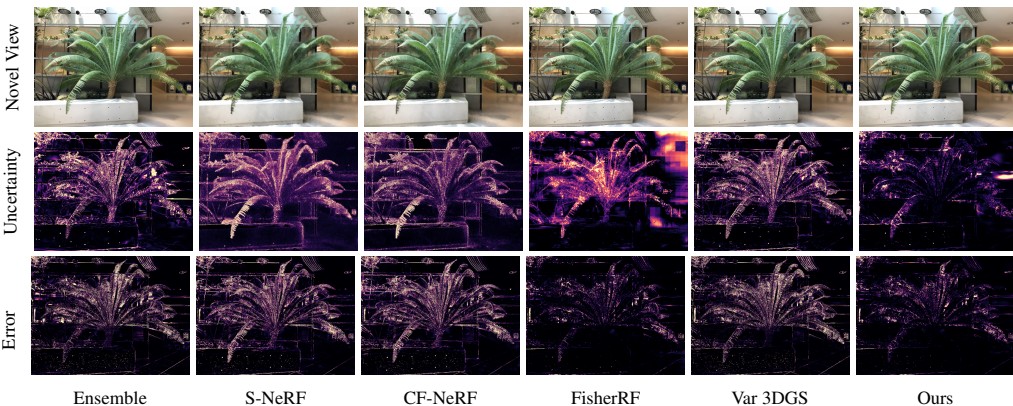

Figure 3: Visualization of predicted uncertainty maps of novel view renderings.

approximately $9\times$ faster than the costly Ensemble GS approach and significantly outpacing both FisherRF and Variational 3DGS. This result demonstrates that our approach provides state-of-the-art performance while simultaneously achieving the fastest rendering speed.

Table 4: The experiment on active learning.

| | PSNR ↑ | SSIM ↑ | LPIPS ↓ |
|---|---|---|---|
| Random | 20.97 | 0.65 | 0.234 |
| FisherRF | 23.37 | 0.81 | 0.144 |
| Variational 3DGS | 21.35 | 0.69 | 0.212 |
| Horseshoe Splatting (Ours) | **26.23** | **0.87** | **0.104** |

Table 5: Inference time for 3DGS-base methods.

| | Inference Time (s) |
|---|---|
| Ensemble GS ($\times 10$) | $0.27 \pm 0.050$ |
| FisherRF | $0.12 \pm 0.003$ |
| Variational 3DGS | $0.06 \pm 0.020$ |
| Horseshoe Splatting (Ours) | $\mathbf{0.03} \pm 0.005$ |

### 5.3 ABLATION STUDY.

**Effectiveness of the Horseshoe Prior.**

To validate our core contribution, we replace the hierarchical Horseshoe prior with a standard Laplace prior and Gaussian prior. As shown in Table 3, while rendering quality remains comparable, the Horseshoe prior yields dramatically better uncertainty estimates. The most striking result is the NLL score, which plummets on both datasets (e.g., from 9.15 to -0.74 on LF), indicating a far more accurate predictive distribution. This confirms the Horseshoe prior's heavy-tailed nature is critical for modeling structural sparsity and producing a well-calibrated predictive distribution.

Table 6: Ablation study on hyperparameters $\rho$ and $\lambda_{\mathrm{KL}}$.

| | Synthesized View Quality | | | Uncertainty Quality | |
|---|---|---|---|---|---|
| | PSNR ↑ | SSIM ↑ | LPIPS ↓ | AUSE ↓ | NLL ↓ |
| $\rho = -5$ | 25.86 | 0.864 | 0.110 | **0.31** | **0.14** |
| $\rho = -4$ | 25.82 | 0.863 | 0.111 | 0.33 | 0.15 |
| $\rho = -3$ | 25.78 | 0.863 | 0.110 | 0.33 | **0.14** |
| $\rho = -2$ | **25.96** | **0.866** | **0.109** | 0.33 | 0.15 |
| $\rho = -1$ | 25.49 | 0.855 | 0.118 | 0.34 | 0.18 |
| $\lambda_{\mathrm{KL}} = 0.1$ | 25.61 | 0.861 | 0.113 | 0.32 | 0.16 |
| $\lambda_{\mathrm{KL}} = 0.01$ | 25.70 | 0.860 | 0.114 | 0.33 | 0.15 |
| $\lambda_{\mathrm{KL}} = 0.001$ | **25.86** | **0.864** | **0.110** | **0.31** | **0.14** |
| $\lambda_{\mathrm{KL}} = 0.0001$ | 25.75 | 0.862 | 0.113 | 0.33 | 0.16 |

**Sensitivity to Hyperparameter Prior Variance $\rho$ and KL Weight $\lambda_{\mathrm{KL}}$.** We study the impact of two key hyperparameters: the log-scale prior variance $\rho$ and the KL-divergence weight $\lambda_{\mathrm{KL}}$, with results on the LLFF dataset shown in Table 6. For $\rho$, which controls the global shrinkage strength, we aim to find a balance between rendering quality and uncertainty estimation. As shown in the table, $\rho = -5$ achieves the best uncertainty scores while maintaining strong rendering performance. Although $\rho = -2$ yields a slightly higher PSNR, its uncertainty metrics are worse. Therefore, to

prioritize well-calibrated uncertainty without significantly compromising rendering quality, we select $\rho = -5$ for our main experiments.

For $\lambda_{\mathrm{KL}}$, the model achieves the best overall performance at $\lambda_{\mathrm{KL}} = 0.001$, leading in both rendering quality and uncertainty metrics. This indicates that while the model is robust, a smaller KL weight is beneficial. We use $\lambda_{\mathrm{KL}} = 0.001$ in our experiments.

**Verification of Structural Sparsity.** Table 7 quantitatively confirms the induced structural sparsity. Across all thresholds $\epsilon \in \{5, 3, 1, 0.5\}$, the Horseshoe prior consistently yields the highest proportion of near-zero scales (e.g., 64.62% vs. 61.01% for Laplace at $\epsilon = 5$). This quantitative evidence is further corroborated by the visualization of the posterior densities in Figure 4. Compared to the Gaussian and Laplace baselines, the Horseshoe posterior over the scale

Table 7: Comparison of Structural Sparsity Induction on LF `torch`.

| Method | Sparsity Ratio (% of scales $< \epsilon$) | | | |
|---|---|---|---|---|
| | $\epsilon = 5$ | $\epsilon = 3$ | $\epsilon = 1$ | $\epsilon = 0.5$ |
| No Prior | 60.73 | 31.57 | 6.59 | 2.98 |
| Gaussian Prior | 60.85 | 31.64 | 6.64 | 3.00 |
| Laplace Prior | 61.01 | 31.95 | 6.78 | 3.07 |
| **Horseshoe Prior** | **64.62** | **34.09** | **7.17** | **3.21** |

parameters $s_{ij}$ exhibits a distinct "spike-at-zero" distribution combined with heavy tails. This behavior indicates aggressive shrinkage of noise-dominated scales while simultaneously preserving large, signal-bearing coefficients—aligning perfectly with the theoretical properties of Horseshoe priors (Carvalho et al., 2009; Piironen and Vehtari, 2017; van der Pas et al., 2017). In contrast, the Laplace prior's lighter tails risk over-shrinking significant features, while the Gaussian prior fails to induce meaningful sparsity. These results confirm that the performance gains are specific to the Horseshoe hierarchy, rather than simply the result of applying "any sparsity prior" (Park and Casella, 2008; Bhadra et al., 2019).

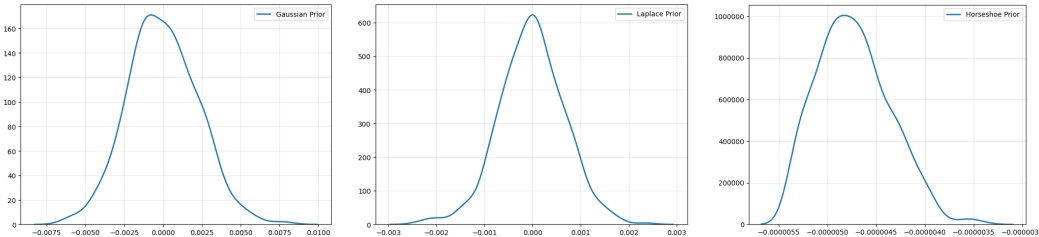

Figure 4: Posterior density of covariance scale ($s_{ij}$) on the LF `torch` scene under different priors (Left: Gaussian; Middle: Laplace; Right: Horseshoe). The Horseshoe prior exhibits a pronounced spike at zero along with heavy tails, inducing structural sparsity by suppressing noise while preserving salient scales, whereas Gaussian and Laplace priors fail to achieve this optimal balance.

## 6 CONCLUSION

We propose Horseshoe Splatting, a Bayesian extension of 3D Gaussian Splatting that places a global–local Horseshoe prior on per-splat covariance scales to encode structural sparsity and deliver calibrated uncertainty. This prior's spike-at-zero and heavy tails aggressively shrink noise-dominated directions while preserving data-supported anisotropy, directly tackling artifacts that arise in deterministic 3DGS pipelines and enabling pixel-wise uncertainty via factorized variational inference and Monte-Carlo rendering. Theoretically, we establish posterior contraction for the scale vector and transfer it to image space under a locally Lipschitz renderer, while empirically we maintain the strong real-time fidelity of 3DGS and obtain reliable uncertainty maps that highlight under-constrained regions. Together, these results show that principled prior design turns explicit splatting into a robust, uncertainty-aware renderer. Looking ahead, we plan to explore alternative structured priors (e.g., low-rank/Wishart constraints) and adaptive splat birth-pruning to further couple sparsity, fidelity, and confidence.

ACKNOWLEDGEMENTS

This work was supported in part by the Research Grants Council of Hong Kong (27206123, 17200125, C5055-24G, and T45-401/22-N), and the Hong Kong Innovation and Technology Fund (GHP/318/22GD).

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

APPENDIX SUMMARY

Section A describes our use of Large Language Models (LLMs). Section B outlines the resources provided to ensure the reproducibility of our work. Section C contains proofs of our main theorems and derivations for the KL divergences. Section D presents additional qualitative results. Section E provides the pseudo-code for our method. Section F provides the additional experiment results.

## A  THE USE OF LARGE LANGUAGE MODELS (LLMS)

Large Language Models (LLMs) were used in this work solely for language polishing and improving the clarity of writing. No LLMs contributed to research ideation, experimental design, analysis, or the development of core scientific content. All conceptual and technical contributions are original to the authors.

## B  REPRODUCIBILITY STATEMENT

To ensure the reproducibility of our work, we provide a comprehensive set of resources. Our full implementation is available as an anonymous code repository, linked in the abstract, which includes all necessary code and usage instructions to replicate our experiments.

## C  TECHNICAL DETAILS

### C.1  PROOFS OF MAIN THEOREMS

*Proof of Lemma 1.* Set $z = J_*(s - s^*)$. Since $\sigma_{\min}(J_*) \geq \kappa_{\min}$, the Moore–Penrose bound gives

$$\|s - s^*\|_2 \leq \|J_*^\dagger\|_{\mathrm{op}} \|z\|_2 = \kappa_{\min}^{-1} \|z\|_2,$$

where $J_*^\dagger$ denotes the Moore–Penrose pseudoinverse of $J_*$. Under the surrogate linear model, the likelihood in $z$ is Gaussian,

$$y = g(s^*) + z + \tilde{\varepsilon}, \qquad \tilde{\varepsilon} \sim \mathcal{N}\left(0, \frac{\sigma^2}{P} I_{p_y}\right).$$

Equipping $s$ with the Horseshoe prior (specified by Assumption 3) induces a global–local shrinkage prior on $z$ through the linear map $J_*$. By the testing–plus–prior–mass program for posterior contraction (Ghosal et al., 2000) together with near–minimax Horseshoe rates in sparse normal means (van der Pas et al., 2014; 2017),

$$\Pi\left(\|z\|_2^2 > C \frac{\sigma^2}{P} k \log \frac{e\,3N}{k} \;\middle|\; y\right) \to 0.$$

Pulling back through $J_*^\dagger$ scales Euclidean error by at most $\kappa_{\min}^{-1}$, hence

$$\|s - s^*\|_2^2 \leq \kappa_{\min}^{-2} \|z\|_2^2 = O_\Pi\left(\frac{\sigma^2}{P} \frac{k \log(e\,3N/k)}{\kappa_{\min}^2}\right),$$

which is the desired rate; the posterior–mean bound follows from Jensen's inequality using the posterior tail control (Ghosal et al., 2000). □

*Proof of Theorem 1.* By a second–order Taylor expansion at $s^*$,

$$y = g(s^*) + J_*(s - s^*) + r(s) + \varepsilon, \qquad \|r(s)\|_2 \leq \frac{H_g}{2} \|s - s^*\|_2^2.$$

For the linearized likelihood with mean $g(s^*) + J_*(s - s^*)$, Lemma 1 (invoking (Ghosal et al., 2000; van der Pas et al., 2014; 2017)) yields contraction at radius

$$\varepsilon_{N,P}^2 = C \frac{\sigma^2}{P} \frac{k \log(e\,3N/k)}{\kappa_{\min}^2}.$$

On the complement of the contraction ball $\{\|\boldsymbol{s} - \boldsymbol{s}^*\|_2 > \varepsilon_{N,P}\}$, the remainder satisfies $\|r(\boldsymbol{s})\|_2 \le \frac{H_g}{2}\|\boldsymbol{s} - \boldsymbol{s}^*\|_2^2 = o(\kappa_{\min}\|\boldsymbol{s} - \boldsymbol{s}^*\|_2)$ under the side condition $H_g\,\varepsilon_{N,P} = o(\kappa_{\min})$. Therefore, the nonlinear and linearized log-likelihood ratios differ by $o(1)$ uniformly on the alternative, so exponentially powerful tests and KL–ball prior mass transfer by Le Cam–type contiguity/bracketing arguments (Ghosal et al., 2000; Nickl, 2022). Consequently,

$$\Pi\!\Big(\|\boldsymbol{s} - \boldsymbol{s}^*\|_2^2 > M\,\varepsilon_{N,P}^2 \;\Big|\; y\Big) \to 0,$$

and $\mathbb{E}_{\boldsymbol{s}^*}\|\mathbb{E}[\boldsymbol{s} \mid y] - \boldsymbol{s}^*\|_2^2 = O(\varepsilon_{N,P}^2)$. $\qquad\square$

*Proof of Theorem 2.* By local Lipschitzness of the renderer (Assumption 5),

$$\|\boldsymbol{I} - \boldsymbol{I}^*\|_2 \le L_{\mathcal{R}}\,\|\boldsymbol{s} - \boldsymbol{s}^*\|_2.$$

Hence for any $M' > 0$,

$$\big\{\|\boldsymbol{I} - \boldsymbol{I}^*\|_2 > M'L_{\mathcal{R}}\varepsilon_{N,P}\big\} \subseteq \big\{\|\boldsymbol{s} - \boldsymbol{s}^*\|_2 > M'\varepsilon_{N,P}\big\}.$$

Taking posterior probabilities and expectations under $\boldsymbol{s}^*$ and applying Theorem 1 gives contraction in image space at rate $\tilde{\varepsilon}_{N,P} = L_{\mathcal{R}}\varepsilon_{N,P}$; this is the standard Lipschitz pushforward used for nonlinear Bayesian inverse problems (Nickl, 2022). $\qquad\square$

## C.2 Closed Forms of KL Divergences

**KL Divergence Between Two Inverse Gamma Distributions** Below is the KL divergence between two inverse-Gamma distributions,

$$q(x) = \text{Inv-}\Gamma(x;\,\alpha_q, \beta_q), \quad p(x) = \text{Inv-}\Gamma(x;\,\alpha_p, \beta_p),$$

where

$$\text{Inv-}\Gamma(x;\,\alpha, \beta) = \frac{\beta^\alpha}{\Gamma(\alpha)}\,x^{-\alpha-1}\exp\!\Big(-\tfrac{\beta}{x}\Big).$$

The KL divergence $\text{KL}(q\|p)$ is

$$\begin{aligned}
\text{KL}(q\|p) &= \int_0^\infty q(x)\,\ln\frac{q(x)}{p(x)}\,dx \\
&= \big(\alpha_q - \alpha_p\big)\,\psi(\alpha_q) - \ln\Gamma(\alpha_q) + \ln\Gamma(\alpha_p) + \alpha_p\ln\frac{\beta_q}{\beta_p} + \alpha_q\frac{\beta_p}{\beta_q} - \alpha_q,
\end{aligned}$$

where $\psi(\cdot)$ is the digamma function.

# D  Additional Visualization Results

## D.1 Depth and Uncertainty on the LF Dataset

Figure 5 shows additional qualitative results for depth rendering and uncertainty estimation on the LF dataset. Our method accurately captures the geometry and produces uncertainty maps that correlate well with the error maps.

## D.2 RGB and Uncertainty on the LLFF Dataset

Figure 6 presents additional qualitative results for novel view synthesis and uncertainty estimation on the LLFF dataset. Across all scenes, our method generates high-fidelity renderings, and the corresponding uncertainty maps correlate well with the reconstruction errors.

# E  Algorithm Details

The pseudo-code for our Horseshoe Splatting algorithm is provided below.

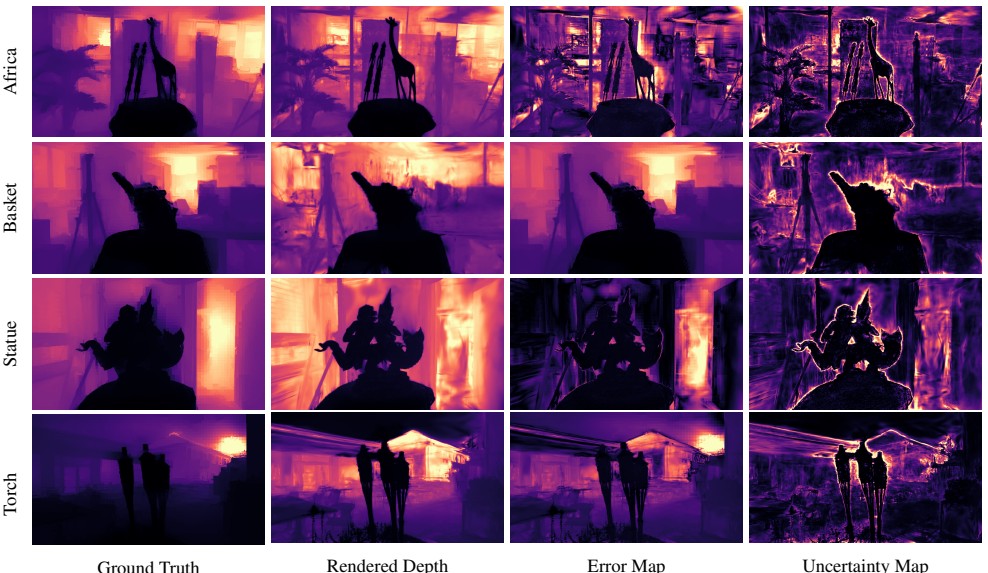

Figure 5: The visualization of depth rendering and uncertainty on LF dataset.

---

**Algorithm 1** Horseshoe Splatting (training and posterior predictive rendering)

---

**Input:** images $D$, camera poses, number of Gaussians $N$, initial $\{\mu_i, R_i, c_i, \alpha_i\}_{i=1}^N$, base scale params $\{\beta_{ij}, \sigma_{ij}\}_{i=1..N, j=1..3}$

**Prior (global–local Horseshoe):** $s_{ij} \mid \lambda_{ij}, \theta_j \sim \mathcal{N}(\beta_{ij}, \sigma_{ij}^2 \theta_j^2 \lambda_{ij}^2), \quad \lambda_{ij}^2 \mid \nu_{ij} \sim \mathrm{IG}(\frac{1}{2}, 1/\nu_{ij}), \ \nu_{ij} \sim \mathrm{IG}(\frac{1}{2}, 1), \quad \theta_j^2 \mid \xi_j \sim \mathrm{IG}(\frac{1}{2}, 1/\xi_j), \ \xi_j \sim \mathrm{IG}(\frac{1}{2}, 1/b^2)$

**Init VI factors:** for all $i, j$ set $q_{\mathrm{IG}}(\lambda_{ij}^2; a_{ij}, b_{ij}), q_{\mathrm{IG}}(\nu_{ij}; c_{ij}, d_{ij})$; for each axis $j$ set $q_{\mathrm{IG}}(\theta_j^2; \alpha_j, \beta_j), q_{\mathrm{IG}}(\xi_j; \gamma_j, \delta_j)$

1: **for** epoch $= 1$ to $n_{\mathrm{epoch}}$ **do**
2:      Sample shrinkage scales: $\lambda_{ij}^2 \sim q_{\mathrm{IG}}(\lambda_{ij}^2), \ \nu_{ij} \sim q_{\mathrm{IG}}(\nu_{ij}), \ \theta_j^2 \sim q_{\mathrm{IG}}(\theta_j^2), \ \xi_j \sim q_{\mathrm{IG}}(\xi_j)$
3:      Sample noise: $\varepsilon_{ij} \sim \mathcal{N}(0, 1)$ and set $\ \ s_{ij} = \beta_{ij} + \sigma_{ij} \theta_j \lambda_{ij} \varepsilon_{ij}$
4:      Build $S_i = \mathrm{diag}(s_{i1}, s_{i2}, s_{i3})$ and $\Sigma_i = R_i S_i R_i^\top$
5:      Render predicted pixels $\{\hat{I}_u\}$ with 3DGS rasterization
6:      $\mathcal{L}_{\mathrm{rec}} \leftarrow -\sum_u \log p(I_u \mid \hat{I}_u)$
7:      $\mathcal{L}_{\mathrm{KL}} \leftarrow \sum_{i,j} \Big( \mathrm{KL}\big[q_{\mathrm{IG}}(\lambda_{ij}^2) \,\|\, \mathrm{IG}(\frac{1}{2}, 1/\nu_{ij})\big] + \mathrm{KL}\big[q_{\mathrm{IG}}(\nu_{ij}) \,\|\, \mathrm{IG}(\frac{1}{2}, 1)\big] \Big)$
8:          $+ \sum_j \Big( \mathrm{KL}\big[q_{\mathrm{IG}}(\theta_j^2) \,\|\, \mathrm{IG}(\frac{1}{2}, 1/\xi_j)\big] + \mathrm{KL}\big[q_{\mathrm{IG}}(\xi_j) \,\|\, \mathrm{IG}(\frac{1}{2}, 1/b^2)\big] \Big)$
9:      Update $\{\mu_i, R_i, c_i, \alpha_i, \beta_{ij}, \sigma_{ij}\}$ and all VI parameters by gradients of $\mathcal{L}_{\mathrm{rec}} + \mathcal{L}_{\mathrm{KL}}$
10: **end for**
     **Posterior predictive & uncertainty (MC rendering):**
11: **for** $m = 1$ to $M$ **do**
12:      Draw $\lambda_{ij}^{2(m)} \sim q_{\mathrm{IG}}(\lambda_{ij}^2), \ \theta_j^{2(m)} \sim q_{\mathrm{IG}}(\theta_j^2), \ \varepsilon_{ij}^{(m)} \sim \mathcal{N}(0, 1)$
13:      $s_{ij}^{(m)} = \beta_{ij} + \sigma_{ij} \theta_j^{(m)} \lambda_{ij}^{(m)} \varepsilon_{ij}^{(m)}, \quad \Sigma_i^{(m)} = R_i \mathrm{diag}(s_{i1}^{(m)}, s_{i2}^{(m)}, s_{i3}^{(m)}) R_i^\top$
14:      Render image $\hat{I}^{(m)}$; store per-pixel samples
15: **end for**
16: **return** posterior mean $\bar{I} = \frac{1}{M} \sum_m \hat{I}^{(m)}$ and per-pixel variance $\mathrm{Var}[\hat{I}^{(m)}]$

---

# F    ADDITIONAL EXPERIMENTAL RESULTS

## F.1    EVALUATION ON CHALLENGING LARGE-SCALE DATASETS

To validate the robustness of our method in diverse and challenging scenarios, we conducted comprehensive additional experiments on the Tanks & Temples and Mip-NeRF 360 datasets, which feature

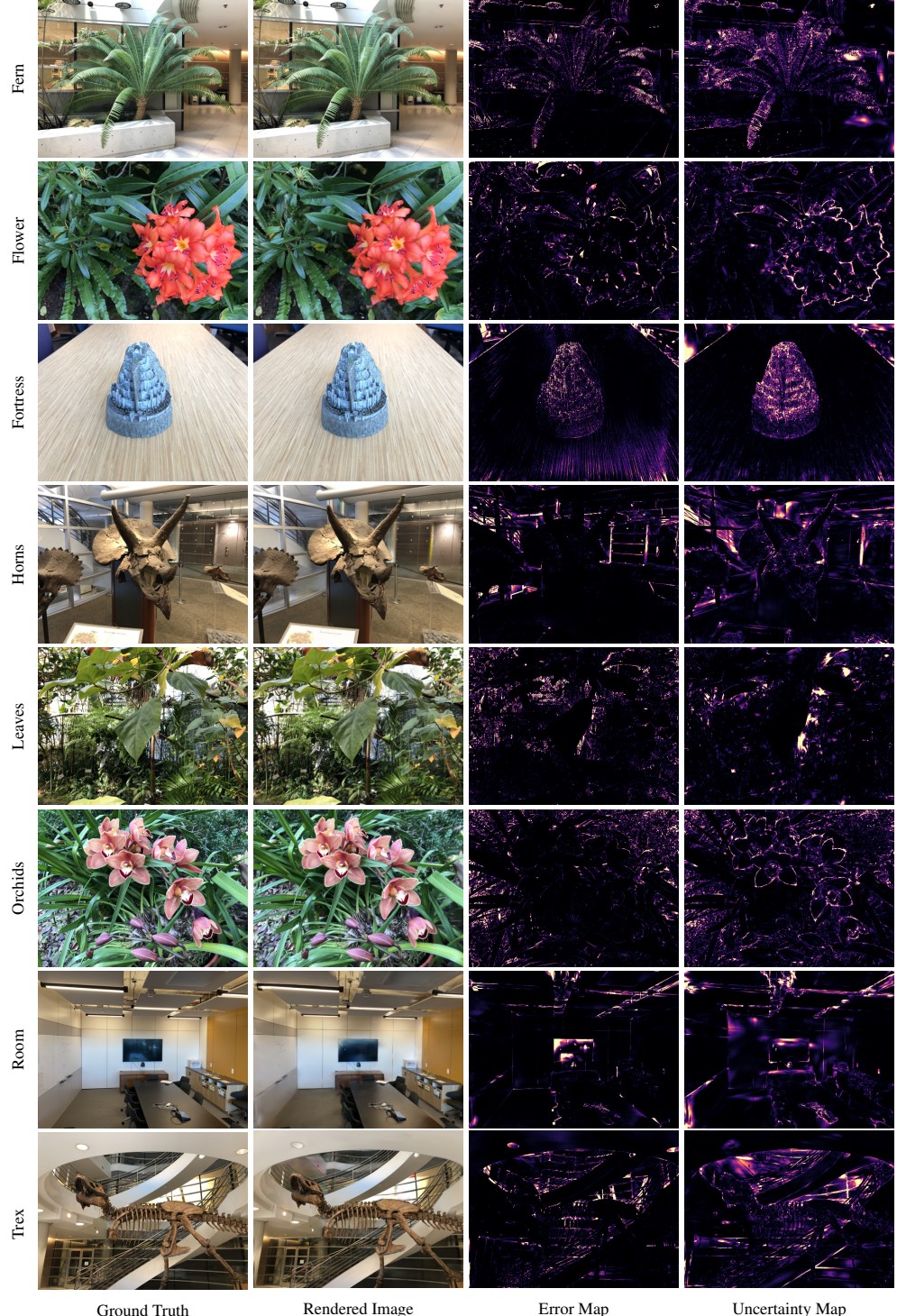

Figure 6: The visualization of RGB images rendering and uncertainty on the LLFF dataset.

large-scale, unbounded outdoor scenes with complex geometries and varying lighting conditions. The quantitative results, presented in Table 8, demonstrate our method's exceptional generalization capabilities. Specifically, we achieve a substantial improvement in uncertainty calibration compared to baselines: on Tanks & Temples, our method reduces the Negative Log-Likelihood (NLL) from 2.46 (Variational 3DGS) to 0.58, and on Mip-NeRF 360 from 2.88 to 0.72. This confirms that

our Horseshoe prior effectively induces meaningful structural sparsity even in complex wild settings. Furthermore, we maintain or exceed the rendering quality (PSNR/SSIM/LPIPS) of competing uncertainty-aware methods, proving that our improved uncertainty modeling does not compromise visual fidelity.

Table 8: Novel View Synthesis (NVS) and Uncertainty Estimation on large-scale datasets (Tanks & Temples and Mip-NeRF360). Our method achieves significantly better uncertainty metrics (AUSE, NLL) while maintaining superior visual quality. Best results are in **boldface**.

| Dataset | Method | Synthesized View Quality | | | Uncertainty Quality | |
|---|---|---|---|---|---|---|
| | | PSNR ↑ | SSIM ↑ | LPIPS ↓ | AUSE ↓ | NLL ↓ |
| Tanks & Temples | FisherRF | 23.41 | 0.837 | 0.198 | 0.57 | 5.94 |
| | Variational 3DGS | 23.45 | 0.835 | 0.199 | 0.47 | 2.46 |
| | **Ours** | **23.67** | **0.845** | **0.186** | **0.35** | **0.58** |
| Mip-NeRF360 | FisherRF | 27.36 | 0.803 | 0.238 | 0.56 | 6.11 |
| | Variational 3DGS | 27.28 | 0.797 | 0.250 | 0.51 | 2.88 |
| | **Ours** | **27.68** | **0.810** | **0.207** | **0.35** | **0.72** |

## F.2 COMPUTATIONAL COST ANALYSIS

To demonstrate accessibility and efficiency, we benchmarked our method against Vanilla 3DGS and other uncertainty-aware methods on a single NVIDIA RTX 3090 GPU (Table 9). It is crucial to note that Vanilla 3DGS is deterministic and inherently incapable of estimating uncertainty; obtaining uncertainty via a standard Ensemble strategy (e.g., the $\times 10$ baseline) would linearly scale the training time and resource usage by a factor of 10, making it prohibitively expensive. While our Bayesian formulation incurs a moderate overhead compared to a single deterministic 3DGS (training: 42s $\rightarrow$ 87s), this represents a highly efficient trade-off compared to ensembles. Moreover, compared to other single-model uncertainty methods, our approach is the only one to maintain real-time rendering speeds (33 FPS), significantly outperforming FisherRF (8 FPS) and Variational 3DGS (17 FPS). With a peak memory usage of only 1.43 GB and a compact model size of 59 MB, our method remains lightweight enough for deployment on commodity hardware, demonstrating its practicality for real-world applications.

Table 9: Computational Cost on the LF `torch` scene. All experiments were conducted on a single NVIDIA RTX 3090 GPU.

| Method | Training Time | FPS | Peak Memory | Model Size |
|---|---|---|---|---|
| 3DGS($\times$ 1) | 42.17s | 494 | 0.47GB | 49MB |
| FisherRF | 55.22s | 8 | 1.20GB | 128MB |
| Variational 3DGS | 81.08s | 17 | 0.87GB | 99MB |
| Ours | 87.33s | 33 | 1.43GB | 59MB |

## F.3 DOWNSTREAM UTILITY OF UNCERTAINTY

To further demonstrate the practical value of our uncertainty estimates, we evaluated our method on two downstream tasks: Active View Selection on the large-scale Tanks & Temples dataset, and Out-of-Distribution (OOD) View Detection on the LLFF dataset. The results are presented in Tables 4 and 11. Regarding active view selection, our uncertainty-guided selection on Tanks & Temples achieves a PSNR of 20.05, significantly outperforming Random selection (16.87) and Variational 3DGS (19.86), confirming that our uncertainty estimates successfully identify the most informative geometric regions under sparse data conditions. In the OOD view detection experiment on LLFF, our method achieves a remarkable AUROC of 0.8732 in identifying novel views, far surpassing Variational 3DGS (0.7345) and FisherRF (0.5696). This demonstrates that our uncertainty is well-calibrated enough to serve as a reliable signal for anomaly detection in safety-critical scenarios.

Table 10: Active Learning on Tanks & Temples.

| Method | PSNR ↑ | SSIM ↑ | LPIPS ↓ |
|---|---|---|---|
| Random | 16.87 | 0.61 | 0.345 |
| FisherRF | 17.00 | 0.61 | 0.346 |
| Variational 3DGS | 19.86 | 0.67 | 0.330 |
| Ours | **20.05** | **0.71** | **0.322** |

Table 11: OOD View Detection on LLFF.

| Method | AUROC |
|---|---|
| FisherRF | 0.5696 |
| Variational 3DGS | 0.7345 |
| Ours | **0.8732** |

