# OpenReview forum: "Horseshoe Splatting: Handling Structural Sparsity for Uncertainty-Aware Gaussian-Splatting Radiance Field Rendering"
_ICLR.cc/2026/Conference — ICLR 2026 Poster_

### Official Review · Reviewer_W9qb · 2025-10-27

**Soundness:** 2
**Presentation:** 3
**Contribution:** 2
**Rating:** 6
**Confidence:** 3

**Summary:**

This paper proposes Horseshoe Splatting, a Bayesian extension of 3DGS for radiance field rendering. The key idea is to introduce a global–local Horseshoe prior over per-splat covariance scales, allowing the model to encode structured sparsity in the Gaussian footprints while also providing pixel-wise uncertainty estimates.  Empirically, the proposed method achieves state-of-the-art image quality and well-calibrated uncertainty on the LF and LLFF benchmarks, outperforming both prior NeRF- and 3DGS-based uncertainty modeling methods, while maintaining real-time rendering speed.

**Strengths:**

- The paper introduces a conceptually clear global–local Horseshoe for 3DGS uncertainty estimation while keeping sufficient efficiency.

- Theoretical derivations are provided to support the method.

- Strong performance is shown on LF and LLFF.

**Weaknesses:**

- It is unclear whether the observed benefits are specific to the Horseshoe, or simply due to introducing any sparsity prior. Also, the ablation only reports improved metrics on the datasets, not how much structural sparsity was actually induced. Thus, the paper lacks quantitative evidence that the prior truly enforces the claimed structural sparsity. It should be further verified whether Horseshoe is necessary and essential.

- The assumption of scaling sparsity places priors on covariance scales, which has not been well verified whether this is sufficiently generalizable for various scenes. Also, as it does not rely on color or opacity, the effectiveness may be limited in scenes dominated by photometric ambiguity rather than geometric uncertainty, such as non-Lambertian surfaces.

- The experiments regarding active view selection may be too ideal. Considering there initially does not contain too many views in the LF dataset, starting with 10% of views may be too sparse and seems to be a toy study that can hardly match the real-world application. Would like to see if the method can still work in more scenarios.

**Questions:**

See the weaknesses.

---

> ### Author Response · Authors · 2025-11-23
>
> We appreciate the reviewer’s constructive feedback and are glad you found our approach conceptually clear and efficient. We provide detailed responses to your questions below.
>
> >**[W1]** It is unclear whether the observed benefits are specific to the Horseshoe, or simply due to introducing any sparsity prior. Also, the ablation only reports improved metrics on the datasets, not how much structural sparsity was actually induced. Thus, the paper lacks quantitative evidence that the prior truly enforces the claimed structural sparsity. It should be further verified whether Horseshoe is necessary and essential.
>
> We thank the reviewer for this critical question regarding the verification of structural sparsity. To determine whether the Horseshoe prior is necessary or if "any sparsity prior" suffices, we analyzed the distribution of learned covariance scales by comparing three distinct categories: the Gaussian (non-sparsity prior), the Laplace (non-structured sparsity prior), and our Horseshoe (structured sparsity prior).
>
> **Table R1** presents this quantitative comparison. While the Laplace prior induces some sparsity compared to the non-sparse Gaussian, it is the Horseshoe prior that consistently yields the highest fraction of near-zero scales across all thresholds $\epsilon \in \{5,3,1,0.5\}$(e.g., 64.62% vs. 61.01% for Laplace). This confirms that the structured sparsity induced by the Horseshoe’s global-local hierarchy is more effective than the simple sparsity of the Laplace. Furthermore, **Figure 4 (in the revised manuscript)** visualizes the mechanism: unlike the Gaussian or the light-tailed Laplace, the Horseshoe posterior exhibits a sharp "spike-at-zero" combined with heavy tails. This allows for aggressive shrinkage of noise (sparsity) without over-shrinking large, salient scales (structure), a property that non-structured priors like Laplace fail to achieve [1–3]. Thus, the benefits are indeed specific to the Horseshoe prior and not merely due to introducing any sparsity prior [4,5].
>
> **Table R1: The experiment on prior choice for structural sparsity on LF `torch`.**
> | Method | Sparsity (ϵ=5) | Sparsity (ϵ=3) | Sparsity (ϵ=1) | Sparsity (ϵ=0.5) |
> | :--- | :---: | :---: | :---: | :---: |
> | w/o prior | 60.73 | 31.57 | 6.59 | 2.98 |
> | Gaussian prior | 60.85 | 31.64 | 6.64 | 3.00 |
> | Laplace prior | 61.01 | 31.95 | 6.78 | 3.07 |
> | Horseshoe prior | **64.62** | **34.09** | **7.17** | **3.21** |
>
> >[1] Carvalho, Polson, Scott. “Handling Sparsity via the Horseshoe.” AISTATS/PMLR, 2009.
>
> >[2] Piironen, Vehtari. “Sparsity information and regularization in the horseshoe and other shrinkage priors.” EJS, 2017.
>
> >[3] van der Pas, Szabó, van der Vaart. “Adaptive posterior contraction rates for the horseshoe.” EJS, 2017.
>
> >[4] Park, Casella. “The Bayesian Lasso.” JASA, 2008.
>
> >[5] Bhadra, Datta, Polson, Willard. “Lasso Meets Horseshoe: A Survey.” Statistical Science, 2019.

---

> > ### Author Response · Authors · 2025-11-23
> >
> > >**[W2]** The assumption of scaling sparsity places priors on covariance scales, which has not been well verified whether this is sufficiently generalizable for various scenes. Also, as it does not rely on color or opacity, the effectiveness may be limited in scenes dominated by photometric ambiguity rather than geometric uncertainty, such as non-Lambertian surfaces.
> >
> > Thanks for your comment. Although not all scene uncertainty stems from its geometric property, the per-pixel contribution of a splat is dominated by its screen-space footprint (projection of the 3D covariance) and the resulting visibility/$\alpha$-compositing. This is exactly what the axis-wise scales control, and why scale uncertainty captures the main ambiguity under sparse views and depth/visibility confusion [6,11]. Our results show that scale-only Bayesian regularization can generalize beyond LF/LLFF: **Table R2** reports strong rendering + calibration on Mip-NeRF360 and Tanks&Temples (outdoor/unbounded scenes), and **Tables R3-R4** show practical gains for active view selection and OOD view detection—use cases where footprint/coverage is the chief driver. Related UQ for NeRFs likewise improves robustness when it models the variables that govern coverage/visibility (e.g., Bayes’ Rays; FisherRF) [7,8].
> >
> > Where photometric ambiguity dominates (strongly non-Lambertian/view-dependent appearance), color/opacity can matter more. This is a limitation of our framework, and we have discussed how to extend our hierarchy in the revised manuscript: (i) band-wise (global–local) Horseshoe on spherical-harmonic color coefficients with stronger shrinkage on higher SH bands to stabilize specular lobes; (ii) logit-normal Horseshoe (or Beta with a global–local scale) on opacity to suppress spurious semi-transparent splats while keeping heavy tails for thin/transparent structures. These choices align with known challenges in 3DGS for view-dependent effects [9,10] and keep the same variational machinery.
> >
> > In summary, our claim is that structural sparsity in the covariance scales is a central, general source of uncertainty in 3DGS, and placing a Horseshoe prior there addresses it in a principled, theoretically grounded way. While color/opacity also contributes, photometric extensions are straightforward within the same framework, and we will include the above validations in future extensions of our work.
> >
> > **Table R2: Novel View Synthesis (NVS) and Uncertainty Estimation on Tanks & Temples and Mip-NeRF360. Our method achieves significantly better uncertainty metrics (AUSE, NLL) while maintaining superior visual quality.**
> > | Dataset | Method | PSNR ↑ | SSIM ↑ | LPIPS ↓ | AUSE ↓ | NLL ↓ |
> > | :--- | :--- | :---: | :---: | :---: | :---: | :---: |
> > | **Tanks&Temples** | FisherRF | 23.41 | 0.837 | 0.198 | 0.57 | 5.94 |
> > | | Variational 3DGS | 23.45 | 0.835 | 0.199 | 0.47 | 2.46 |
> > | | **Ours** | **23.67** | **0.845** | **0.186** | **0.35** | **0.58** |
> > | **Mip-NeRF360** | FisherRF | 27.36 | 0.803 | 0.238 | 0.56 | 6.11 |
> > | | Variational 3DGS | 27.28 | 0.797 | 0.250 | 0.51 | 2.88 |
> > | | **Ours** | **27.68** | **0.810** | **0.207** | **0.35** | **0.72** |
> >
> > >[6] Kerbl et al., “3D Gaussian Splatting for Real-Time Radiance Field Rendering,” 2023.
> >
> > >[7] Goli et al., “Bayes’ Rays: Uncertainty Quantification for Neural Radiance Fields,” CVPR 2024.
> >
> > >[8] Jiang et al., “FisherRF: Active View Selection and Uncertainty Quantification for Radiance Fields,” 2023.
> >
> > >[9] Wilson et al., “Modeling Uncertainty in 3D Gaussian Splatting Through Continuous Semantic Splatting,” 2024/2025.
> >
> > >[10] “High-quality Modeling of the View-dependent Appearance for 3D Gaussian Splatting,” 2024. (challenges with non-Lambertian/specular appearance in 3DGS).
> >
> > >[11] C Chen and W Wang. Survey: “A Survey on 3D Gaussian Splatting,” 2025 (footprint/SH overview).

---

> > > ### Author Response · Authors · 2025-11-23
> > >
> > > >**[W3]** The experiments regarding active view selection may be too ideal. Considering there initially does not contain too many views in the LF dataset, starting with 10\% of views may be too sparse and seems to be a toy study that can hardly match the real-world application. Would like to see if the method can still work in more scenarios.
> > >
> > > We thank the reviewer for the comment regarding our experimental setting.
> > > First, we would like to clarify that **our original active view selection experiment was conducted on the LLFF dataset**, not the LF dataset. While LF is indeed a structured grid, LLFF features unstructured, forward-facing camera poses with sparse inputs. Indeed, LLFF is a standard benchmark for active view selection utilized in prior works [12,13], chosen precisely because it approximates real-world handheld capture better than grid datasets.
> > > Nevertheless, we fully agree with the reviewer that validating on broader, more complex scenarios is essential to prove that our method is not a "toy study." To address this, we have conducted two new experiments in active learning and OOD view detection.
> > >
> > > First, we scaled our active view selection experiment to the Tanks&Temples dataset, a large-scale outdoor scene with complex geometry. As shown in **Table R3**, our method achieves a PSNR of 20.05, significantly outperforming the Random baseline (16.87) and Variational 3DGS (19.86). This proves our method works effectively in non-ideal, large-scale environments.
> > > Second, to demonstrate utility in a completely different scenario, we evaluated Out-of-Distribution (OOD) view detection on LLFF. As shown in **Table R4**, our method achieves an AUROC of 0.8732, proving it can robustly identify unsafe/unreliable views in real-world applications.
> > >
> > > **Table R3: The Active Learning Experiment on Tanks&Temples.**
> > > | Method | PSNR ↑ | SSIM ↑ | LPIPS ↓ |
> > > | :--- | :---: | :---: | :---: |
> > > | Random | 16.87 | 0.61 | 0.345 |
> > > | FisherRF | 17.00 | 0.61 | 0.346 |
> > > | Variational 3DGS | 19.86 | 0.67 | 0.330 |
> > > | Ours | **20.05** | **0.71** | **0.322** |
> > >
> > > **Table R4: The OOD View Detection Experiment on LLFF.**
> > > | Method | AUROC |
> > > | :--- | :---: |
> > > | FisherRF | 0.5696 |
> > > | Variational 3DGS | 0.7345 |
> > > | Ours | **0.8732** |
> > >
> > > > [12] Pan et al. "Activenerf: Learning where to see with uncertainty estimation." ECCV, 2022.
> > >
> > > > [13] Li R, Cheung Y. "Variational multi-scale representation for estimating uncertainty in 3d gaussian splatting." NeurIPS, 2024.

---

> ### Comment · Reviewer_W9qb · 2025-11-24
>
> Thanks for the responses. I'm mostly satisfied with the complemented verifications and discussions and would like to see them added in the camera ready. Besides, I'd like to note that in line 431, the description still shows that the active view selection experiment was done on LF dataset, rather than the claimed LLFF in the rebuttal. Please check if it is correct. I'll maintain my positive recommendation.

---

> > ### Author Response · Authors · 2025-11-24
> >
> > We sincerely thank you for your prompt response and for maintaining your positive recommendation.
> >
> > You are absolutely correct. The mention of "LF" in line 431 is indeed a typo; it should be LLFF, consistent with our experimental setup and rebuttal claims. We apologize for this confusion and have corrected this error in the revised manuscript.

---

### Official Review · Reviewer_QC4h · 2025-10-30

**Soundness:** 3
**Presentation:** 2
**Contribution:** 2
**Rating:** 4
**Confidence:** 4

**Summary:**

This paper introduces Horseshoe Splatting, a Bayesian variant of 3D Gaussian Splatting (3DGS) that explicitly models structural sparsity and uncertainty in radiance field rendering. The method places a global–local Horseshoe prior on per-splat covariance scales to suppress noisy or redundant directions while retaining key anisotropic structures. A variational inference framework enables Monte Carlo rendering and pixel-wise uncertainty estimation with little computational cost. Experiments on Light Field and LLFF datasets show that the approach achieves state-of-the-art image quality and well-calibrated uncertainty, while maintaining real-time rendering performance.

**Strengths:**

1. Innovative Bayesian extension of 3DGS that meaningfully incorporates structural sparsity and uncertainty estimation.
  2. Solid theoretical grounding, with posterior contraction guarantees linking model uncertainty to data consistency.
  3. Strong empirical validation, outperforming prior methods on fidelity and uncertainty metrics without slowing inference.
  4. Clear writing and presentation, with convincing visual and quantitative results.

**Weaknesses:**

1. Limited Conceptual Novelty Beyond Integration. While the Horseshoe prior is applied creatively to 3DGS, the overall contribution is more of a principled integration of known Bayesian shrinkage techniques rather than a fundamentally new rendering paradigm. The work’s novelty thus lies mainly in adaptation, not invention.
  2. Dependence on Heavy Computational Infrastructure. The approach relies on large-scale Gaussian Splatting models and variational inference, which may limit reproducibility and accessibility. It’s unclear how well the method scales down to smaller datasets or lightweight hardware settings.
  3. Lack of Diversity in Experimental Scenarios. Experiments are limited to static and relatively clean datasets (LF, LLFF). The paper does not demonstrate how the model behaves under dynamic scenes, noisy inputs, or severe view sparsity — conditions where uncertainty modeling would be most critical.
  4. Insufficient Analysis of Practical Utility of Uncertainty. While uncertainty maps are visually convincing, the paper provides little quantitative evidence of how uncertainty benefits downstream tasks, beyond one active view selection experiment. More demonstration of practical value would strengthen the impact.

**Questions:**

1. Generalization and Robustness – How does Horseshoe Splatting perform on dynamic or large-scale outdoor scenes where scene sparsity and motion are more complex?
  2. Computational Cost – What is the training time and memory overhead compared to vanilla 3DGS? Could this method realistically run on commodity GPUs?
  3. Ablation on Horseshoe Hierarchy – How sensitive is performance to the hierarchical prior choice? Would simpler priors (e.g., Laplace or Gaussian scale mixtures) achieve similar uncertainty quality?
  4. Downstream Use of Uncertainty – Beyond visualization, have the authors tested whether the uncertainty maps improve robustness in active learning or out-of-distribution detection?
  5. Scalability and Model Release – Are there plans to release pretrained models or a lighter implementation for reproducibility?

---

> ### Author Response · Authors · 2025-11-23
>
> We thank the reviewer for the detailed feedback and for acknowledging our solid theoretical grounding and strong empirical validation. We would like further to clarify the novelty and contribution of our work. Our framework targets the raised concerns here.
>
> >**[W1]** Limited Conceptual Novelty Beyond Integration. While the Horseshoe prior is applied creatively to 3DGS, the overall contribution is more of a principled integration of known Bayesian shrinkage techniques rather than a fundamentally new rendering paradigm. The work’s novelty thus lies mainly in adaptation, not invention.
>
> Thank you for the comments. We would like to further clarify the novelty and contribution of our work. Our work targets an essential gap in 3DGS: structural sparsity in the per-splat covariance and the lack of principled, posterior-based uncertainty. In 3D Gaussian Splatting, the anisotropic covariance is the workhorse parameter that governs the screen-space footprint and visibility weighting; existing pipelines optimize it deterministically and rely on heuristics for density control, leaving redundancy and noise under sparse views insufficiently regulated.
> To alleviate this problem, we introduce the global–local shrinkage prior system to address the structural sparsity challenge and uncertainty estimation. Our choice of a Horseshoe prior is not arbitrary: its spike at zero and heavy tails provide exactly the behavior needed to adaptively zero-out noise-dominated directions while preserving large, data-supported axes—precisely the point highlighted by Reviewer WvZK.
> We are not aware of prior 3DGS works that place a hierarchical shrinkage prior on the covariance scales themselves; related efforts focus on post-hoc pruning via sensitivity (PUP 3D-GS), task-specific probabilistic semantics (CSS), or offset-based VI that samples attributes without structural sparsification, none of which regularize the covariance scales with a Bayesian global–local prior or provide our contraction-type guarantees.
> Our framing, therefore, addresses a central modeling deficiency of 3DGS rather than proposing a different renderer: we regularize the parameter that matters most for footprint/visibility and deliver calibrated, pixel-wise uncertainty with minimal overhead.
>
> >**[W2]&[Q2]** Dependence on Heavy Computational Infrastructure. The approach relies on large-scale Gaussian Splatting models and variational inference, which may limit reproducibility and accessibility. It’s unclear how well the method scales down to smaller datasets or lightweight hardware settings.
> >
> > Computational Cost.What is the training time and memory overhead compared to vanilla 3DGS? Could this method realistically run on commodity GPUs?
>
> We appreciate the reviewer’s comments regarding computational efficiency. We respectfully clarify that our method does not require a heavy computational infrastructure since we employ a factorized variational inference scheme. In fact, it is designed to be lightweight and accessible. To demonstrate this, we benchmarked our method against Vanilla 3DGS and uncertainty baselines on a single consumer-grade NVIDIA RTX 3090 (see **Table R1**).
>
> Specifically, with a peak training memory usage of only 1.43 GB and a compact model size of 59 MB, our method fits easily within the constraints of mid-range or even laptop GPUs, directly addressing concerns about accessibility.
> Regarding scalability to smaller datasets or lightweight settings, we note that  our Bayesian inference scheme incurs a cost proportional to the number of Gaussian primitives; therefore, for smaller datasets that naturally require fewer Gaussians, both the training time and memory footprint decrease linearly, ensuring efficiency.
>
> When comparing with baselines, it is also crucial to note that Vanilla 3DGS is deterministic; achieving uncertainty via a standard Ensemble (e.g., 10 models) would multiply resource usage by 10, making it prohibitively expensive. While our method incurs a moderate training overhead (42s $\to$ 87s) compared to a single deterministic model, it is highly efficient compared to ensembles.
> Finally, regarding inference speed, we represent the only uncertainty-aware method maintaining real-time rendering (33 FPS), significantly outperforming FisherRF (8 FPS) and Variational 3DGS (17 FPS).
>
> **Table R1: Computational Cost on the LF `torch` scene. All experiments were conducted on a single NVIDIA RTX 3090 GPU.**
> | Method | Training Time | FPS | Peak Memory | Model Size |
> | :--- | :---: | :---: | :---: | :---: |
> | 3DGS (×1) | 42.17s | 494 | 0.47GB | 49MB |
> | FisherRF | 55.22s | 8 | 1.20GB | 128MB |
> | Variational 3DGS | 81.08s | 17 | 0.87GB | 99MB |
> | Ours | 87.33s | 33 | 1.43GB | 59MB |

---

> > ### Author Response · Authors · 2025-11-23
> >
> > >**[W3]&[Q1]** Lack of Diversity in Experimental Scenarios. Experiments are limited to static and relatively clean datasets (LF, LLFF). The paper does not demonstrate how the model behaves under dynamic scenes, noisy inputs, or severe view sparsity — conditions where uncertainty modeling would be most critical.
> > >
> > > Generalization and Robustness – How does Horseshoe Splatting perform on dynamic or large-scale outdoor scenes where scene sparsity and motion are more complex?
> >
> > We thank the reviewer for this valuable comment. To demonstrate the robustness of our framework beyond controlled datasets, we have significantly expanded our experimental evaluation to address these specific scenarios:
> >
> > First, to verify generalization to **large-scale outdoor scenes** with complex geometry, we conducted full training and evaluation on the Mip-NeRF360 and Tanks&Temples, which feature unbounded environments and varying lighting. As shown in **Table R2**, our method outperforms baselines in uncertainty calibration (NLL 0.58 vs. 2.46 for baselines), demonstrating that it effectively handles **non-clean** geometry.
> >
> > Second, to assess performance under **severe view sparsity**, we designed an active learning setup initialized with a minimal set of only 10% of views. By iteratively adding views based on predictive uncertainty, we aim to verify if our method can guide reconstruction when data is scarce. As shown in **Table 3** and **Table R3**, our method significantly outperforms random selection and other baselines, confirming that our uncertainty estimates provide a reliable signal for data-efficient reconstruction.
> >
> > Third, to evaluate robustness under **critical conditions**, we established an Out-of-Distribution (OOD) view detection task. We defined test views outside the training camera distribution as OOD and used our uncertainty score to detect them. As shown in **Table R4**, our method achieves an AUROC of 0.8732, proving that it can reliably identify unsafe or unknown operating conditions in practical applications.
> >
> > Finally, regarding dynamic scenes, while our current implementation follows the standard 3DGS formulation for static scenes, the robust performance on complex outdoor geometry suggests our prior is a strong foundation for future 4D extensions.
> >
> > **Table R2: Novel View Synthesis (NVS) and Uncertainty Estimation on Tanks & Temples and Mip-NeRF360. Our method achieves significantly better uncertainty metrics (AUSE, NLL) while maintaining superior visual quality.**
> > | Dataset | Method | PSNR ↑ | SSIM ↑ | LPIPS ↓ | AUSE ↓ | NLL ↓ |
> > | :--- | :--- | :---: | :---: | :---: | :---: | :---: |
> > | **Tanks&Temples** | FisherRF | 23.41 | 0.837 | 0.198 | 0.57 | 5.94 |
> > | | Variational 3DGS | 23.45 | 0.835 | 0.199 | 0.47 | 2.46 |
> > | | **Ours** | **23.67** | **0.845** | **0.186** | **0.35** | **0.58** |
> > | **Mip-NeRF360** | FisherRF | 27.36 | 0.803 | 0.238 | 0.56 | 6.11 |
> > | | Variational 3DGS | 27.28 | 0.797 | 0.250 | 0.51 | 2.88 |
> > | | **Ours** | **27.68** | **0.810** | **0.207** | **0.35** | **0.72** |
> >
> > **Table R3: The Active Learning Experiment on Tanks&Temples.**
> > | Method | PSNR ↑ | SSIM ↑ | LPIPS ↓ |
> > | :--- | :---: | :---: | :---: |
> > | Random | 16.87 | 0.61 | 0.345 |
> > | FisherRF | 17.00 | 0.61 | 0.346 |
> > | Variational 3DGS | 19.86 | 0.67 | 0.330 |
> > | Ours | **20.05** | **0.71** | **0.322** |
> >
> > **Table R4: The OOD View Detection Experiment on LLFF.**
> > | Method | AUROC |
> > | :--- | :---: |
> > | FisherRF | 0.5696 |
> > | Variational 3DGS | 0.7345 |
> > | Ours | **0.8732** |

---

> > > ### Author Response · Authors · 2025-11-23
> > >
> > > >**[W4]&[Q4]** Insufficient Analysis of Practical Utility of Uncertainty. While uncertainty maps are visually convincing, the paper provides little quantitative evidence of how uncertainty benefits downstream tasks, beyond one active view selection experiment. More demonstration of practical value would strengthen the impact.
> > > >
> > > > Downstream Use of Uncertainty – Beyond visualization, have the authors tested whether the uncertainty maps improve robustness in active learning or out-of-distribution detection?
> > >
> > > We appreciate the request for more quantitative evidence. Beyond visualizations, we have demonstrated that our uncertainty map can facilitate two concrete downstream tasks:
> > >
> > > First, regarding active view selection (**Table R3**), we extended our evaluation to the large-scale Tanks & Temples dataset. By using our pixel-wise uncertainty to guide the selection of next-best views, our method yields a PSNR of 20.05, significantly outperforming Random selection (16.87) and the Variational 3DGS baseline (19.86). This quantitative gain aligns with prior work [1–3], confirming that our uncertainty serves as a reliable, high-quality signal for data-efficient reconstruction.
> > >
> > > Furthermore, regarding OOD view detection (**Table R4**), we tested the model's ability to flag unsafe views on the LLFF dataset. Treating our uncertainty as an anomaly score achieves an AUROC of 0.8732, far surpassing FisherRF (0.5696) and Variational 3DGS (0.7345). This demonstrates that our estimates are not only calibrated (cf. AUSE/NLL in **Table 2**) but also useful for robustness to distribution shift, similar in spirit to Bayes’ Rays, which uses NeRF uncertainty to clean up artifacts and detect unreliable regions [4].
> > >
> > > We have clarified these points in the revision by (i) explicitly framing active view selection and OOD detection as practical downstream applications of our uncertainty, and (ii) briefly connecting them to existing NeRF UQ uses in view planning and artifact removal [1–4].
> > >
> > > *[1] Jiang et al., FisherRF: Active View Selection and Uncertainty Quantification for Radiance Fields using Fisher Information, ECCV 2024.*
> > >
> > > *[2] Xue et al., Neural Visibility Field for Uncertainty-Driven Active Mapping, 2024.*
> > >
> > > *[3] Kim et al., Active Neural 3D Reconstruction with Colorized Surface Voxel-based View Selection, 2024.*
> > >
> > > *[4] Goli et al., Bayes’ Rays: Uncertainty Quantification for Neural Radiance Fields, CVPR 2024.*

---

> > > > ### Author Response · Authors · 2025-11-23
> > > >
> > > > >**[Q3]** Ablation on Horseshoe Hierarchy – How sensitive is performance to the hierarchical prior choice? Would simpler priors (e.g., Laplace or Gaussian scale mixtures) achieve similar uncertainty quality?
> > > >
> > > > Thank you for the comments. We have run a prior sweep (**Table R5**): on both LF and LLFF, the Horseshoe prior matches or slightly improves view quality (PSNR/SSIM/LPIPS) and gives clearly better uncertainty (lower AUSE and NLL) than Laplace and Gaussian baselines. This is consistent with known theory: the Horseshoe is a global–local prior with an infinite spike at zero and Cauchy-like heavy tails, which yields aggressive shrinkage of near-zero coefficients while avoiding over-shrinkage of large signals—the exact behavior needed to prune noise-dominated scale components yet preserve true structure in 3DGS covariances. These properties are well documented and underpin near-minimax posterior contraction for sparse signals, explaining why uncertainty estimates become cleaner and better calibrated when using Horseshoe versus simpler priors.
> > > >
> > > > Theoretically speaking, Laplace (L1) has lighter (exponential) tails and is known to over-shrink large effects, which can bias the retained scales and inflate predictive NLL/AUSE; a Gaussian prior is not sparsity-inducing at all (no spike at zero), so it cannot selectively suppress noise near zero. Gaussian scale mixtures such as Normal–Inverse-Gamma (Student-t) do provide heavy tails, but—without a spike at zero—they typically lack the strong near-zero shrinkage that Horseshoe delivers, so they do not remove small, spurious components as effectively. These contrasts are standard in the shrinkage literature [5–10].
> > > >
> > > > **Table R5: Sensitivity Analysis on Prior Choice.**
> > > > | Dataset | Method | PSNR ↑ | SSIM ↑ | LPIPS ↓ | AUSE ↓ | NLL ↓ | Depth AUSE ↓ |
> > > > | :--- | :--- | :---: | :---: | :---: | :---: | :---: | :---: |
> > > > | **LF** | Laplace Prior | 30.04 | 0.942 | 0.065 | 0.37 | 10.58 | 0.19 |
> > > > | | Gaussian Prior | 30.01 | 0.941 | 0.067 | 0.38 | 9.15 | 0.18 |
> > > > | | Horseshoe Prior | **30.05** | **0.947** | **0.064** | **0.25** | **-0.74** | 0.18 |
> > > > | **LLFF** | Laplace Prior | 25.74 | 0.860 | 0.112 | 0.42 | 8.22 | / |
> > > > | | Gaussian Prior | 25.61 | 0.859 | 0.116 | 0.42 | 6.98 | / |
> > > > | | Horseshoe Prior | **25.86** | **0.864** | **0.110** | **0.31** | **0.14** | / |
> > > >
> > > > *[5] Carvalho, Polson, Scott. “Handling Sparsity via the Horseshoe.” AISTATS (PMLR), 2009.*
> > > >
> > > > *[6] Piironen, Vehtari. “Sparsity Information and Regularization in the Horseshoe and Other Shrinkage Priors.” Electron. J. Stat., 2017.*
> > > >
> > > > *[7] van der Pas, Kleijn, van der Vaart. “The Horseshoe Estimator: Posterior Concentration Around Nearly Black Vectors.” Electron. J. Stat., 2014.*
> > > >
> > > > *[8] van der Pas, Szabó, van der Vaart. “Adaptive Posterior Contraction Rates for the Horseshoe.” Electron. J. Stat., 2017.*
> > > >
> > > > *[9] Bhadra et al. “Lasso Meets Horseshoe: A Survey.” Statistical Science, 2019. (Laplace vs. Horseshoe, tail/overshrinkage discussion.)*
> > > >
> > > > *[10] Bhattacharya et al. “Sub-optimality of Some Continuous Shrinkage Priors.” J. Multivariate Analysis, 2016. (Limitations of Bayesian Lasso–type priors.)*
> > > >
> > > > >**[Q5]** Scalability and Model Release – Are there plans to release pretrained models or a lighter implementation for reproducibility?
> > > >
> > > > Thank you for the suggestions. Yes, we have already included the anonymous Github for codes, and will upload the all pretrained models after the paper is accepted for better reproducibility. We now provide some representative models for illustrative purposes.

---

### Official Review · Reviewer_xYVt · 2025-11-01

**Soundness:** 3
**Presentation:** 3
**Contribution:** 3
**Rating:** 6
**Confidence:** 2

**Summary:**

This paper proposes Horseshoe Splatting, which applies a global–local Horseshoe shrinkage prior to the covariance scales of each Gaussian in 3D Gaussian Splatting (3DGS) to encode structured sparsity. Using an inverse-gamma augmentation equivalent to the half-Cauchy prior, the method builds a factorized variational family for Monte Carlo rendering and pixel-wise uncertainty estimation. The authors also provide theoretical guarantees on posterior contraction under Lipschitz assumptions and validate the method on LF/LLFF datasets with improved view and uncertainty quality.

**Strengths:**

1. The paper’s exploration of uncertainty modeling in 3DGS is valuable. Incorporating both uncertainty and structured sparsity is novel, as most existing 3DGS uncertainty modeling focuses on geometry, semantic fields, or Fisher information approximations, with little direct regularization on covariance structures.
2. The experiments are comprehensive, jointly evaluating RGB and depth uncertainty (e.g., NLL, AUSE) and active view selection scenarios, with ablations (Gaussian vs. Horseshoe) demonstrating the benefit of the proposed prior.

**Weaknesses:**

1. The local Lipschitz assumption may be fragile since 3DGS rendering involves depth sorting and α-blending, which can cause discontinuities around occlusion or visibility changes. More clarification or evidence on how local neighborhoods avoid these discrete transitions would strengthen the theory section.
2. The active-view selection strategy lacks details — the paper mentions adding one view every 500 steps until 30%, but the convergence criterion, acquisition function, and budget alignment across methods are not clearly defined.

**Questions:**

It would be helpful if the authors could validate the method on more wild or challenging scenes, such as MipNeRF or Tanks and Temples, to demonstrate broader applicability.

---

> ### Author Response · Authors · 2025-11-23
>
> We thank the reviewer for the positive evaluation and for appreciating the novelty and comprehensive validation of our work. We present our point-to-point responses to the questions below.
>
> >**[W1]** The local Lipschitz assumption may be fragile since 3DGS rendering involves depth sorting and $\alpha$-blending, which can cause discontinuities around occlusion or visibility changes. More clarification or evidence on how local neighborhoods avoid these discrete transitions would strengthen the theory section.
>
> Thank you for the thorough observation and insightful comment. The global smoothness is not guaranteed in our framework. Our theory, however, assumes a local neighborhood around the truth where depth ordering and the **active set** of contributors per pixel are stable. Concretely, we condition on (i) a depth-margin: for all projected centers affecting a pixel,
> $
> \min_{k\neq \ell}\lvert z_k - z_\ell\rvert\ge\gamma>0,
> $
> and (ii) an activity-margin: the set of splats whose screen–space density at that pixel exceeds a tiny threshold is unchanged under
> $
> \lVert \mathbf{s} - \mathbf{s}^{\star} \rVert \le r.
> $
> In our analysis the means $\mu_i$ and orientations $R_i$ are fixed and only the scales $\mathbf{s}$ vary, so these margins imply the sort order does not change and the same contributors are blended; $\alpha$-compositing itself is a smooth composition of sums and exponentials in $\mathbf{s}$. Hence the renderer is $C^1$ (in particular, locally Lipschitz) on that neighborhood, which is all we need for the contraction transfer. Discontinuities can only occur at ordering ties or contributor changes; these events form a measure–zero set under continuous perturbations, and the margins above exclude them in a small ball around $\mathbf{s}^{\star}$.
>
> >**[W2]** The active-view selection strategy lacks details — the paper mentions adding one view every 500 steps until 30\%, but the convergence criterion, acquisition function, and budget alignment across methods are not clearly defined.
>
> We sincerely thank the reviewer for highlighting the need for greater clarity in our active view selection experiment. We have revised the manuscript to explicitly describe our protocol, which follows the setting established in the previous work [1]. The details are as follows:
>
> Our acquisition function is defined to select the view with the highest integrated predictive uncertainty, which is calculated by summing the pixel-wise variance across each candidate image.
> To ensure a rigorous and fair comparison, budget alignment is strictly enforced: our method and all baselines commence with the same initial 10% of views and acquire exactly one new view every 500 steps.
> The process employs a fixed budget stopping criterion, terminating once the training set reaches 30% of the total available views and the model is further trained for 7K steps after all images are chosen.
>
> *[1] Li R, Cheung Y. "Variational multi-scale representation for estimating uncertainty in 3d gaussian splatting." NeurIPS, 2024.*
>
> >**[Q1]** It would be helpful if the authors could validate the method on more wild or challenging scenes, such as MipNeRF or Tanks and Temples, to demonstrate broader applicability.
>
> We thank the reviewer for this suggestion. We agree that validating on large-scale, unbounded scenes is crucial for demonstrating broader applicability. Therefore, we have conducted comprehensive experiments on both the Tanks&Temples and Mip-NeRF360 datasets.
>
> The results are presented in **Table R1**, and it is observed that our method demonstrates strong generalization capabilities.
> First, our method outperforms baselines across all datasets in uncertainty estimation. Notably, on Tanks&Temples, our method reduces the NLL from 2.46 to 0.58, and on Mip-NeRF360 from 2.88 to 0.72. This confirms that our Horseshoe prior can effectively handle structural sparsity even in complex **wild** settings.
> Moreover, we maintain or exceed the rendering quality (PSNR/SSIM/LPIPS) of competing uncertainty-aware methods.
>
> We have included these results in the revised manuscript to demonstrate the broad applicability of our approach.
>
> **Table R1: Novel View Synthesis (NVS) and Uncertainty Estimation on Tanks & Temples and Mip-NeRF360. Our method achieves significantly better uncertainty metrics (AUSE, NLL) while maintaining superior visual quality.**
> | Dataset | Method | PSNR ↑ | SSIM ↑ | LPIPS ↓ | AUSE ↓ | NLL ↓ |
> | :--- | :--- | :---: | :---: | :---: | :---: | :---: |
> | **Tanks&Temples** | FisherRF | 23.41 | 0.837 | 0.198 | 0.57 | 5.94 |
> | | Variational 3DGS | 23.45 | 0.835 | 0.199 | 0.47 | 2.46 |
> | | **Ours** | **23.67** | **0.845** | **0.186** | **0.35** | **0.58** |
> | **Mip-NeRF360** | FisherRF | 27.36 | 0.803 | 0.238 | 0.56 | 6.11 |
> | | Variational 3DGS | 27.28 | 0.797 | 0.250 | 0.51 | 2.88 |
> | | **Ours** | **27.68** | **0.810** | **0.207** | **0.35** | **0.72** |

---

### Official Review · Reviewer_WvZK · 2025-11-01

**Soundness:** 3
**Presentation:** 3
**Contribution:** 3
**Rating:** 6
**Confidence:** 3

**Summary:**

This paper introduces Horseshoe Splatting, a Bayesian extension of 3D Gaussian Splatting (3DGS) that addresses structural sparsity in per-splat covariances while providing calibrated uncertainty estimates. The key innovation is the application of a global-local Horseshoe prior on covariance scales, which adaptively shrinks noise-dominated directions while preserving salient anisotropic structure. The authors develop a factorized variational inference scheme that enables Monte Carlo rendering with pixel-wise posterior uncertainty estimation. Theoretically, they establish posterior contraction rates for scale parameters and propagate these guarantees to rendered images via local Lipschitz mapping. Empirically, the method achieves state-of-the-art visual fidelity on standard benchmarks while producing high-quality uncertainty maps with minimal computational overhead.

**Strengths:**

- The paper addresses a genuine limitation of existing 3DGS methods: the lack of explicit structural sparsity encoding in per-splat covariances and the absence of principled uncertainty quantification
- The observation that noise-dominated components in covariance structures remain insufficiently regularized is well-articulated and demonstrated through visualizations (Figure 1)
- The choice of Horseshoe prior is well-justified by its spike-at-zero and heavy-tail properties, which are precisely what is needed for adaptive sparsification
- The method achieves state-of-the-art uncertainty estimation (Table 1: average AUSE of 0.18 on depth, NLL of -0.74 on RGB for LF dataset)
- Novel view synthesis quality is not sacrificed for uncertainty: PSNR of 30.05 on LF dataset, outperforming baselines by significant margins

**Weaknesses:**

- The paper addresses a genuine limitation of existing 3DGS methods: the lack of explicit structural sparsity encoding in per-splat covariances and the absence of principled uncertainty quantification
- The observation that noise-dominated components in covariance structures remain insufficiently regularized is well-articulated and demonstrated through visualizations (Figure 1)
- The choice of Horseshoe prior is well-justified by its spike-at-zero and heavy-tail properties, which are precisely what is needed for adaptive sparsification
- The method achieves state-of-the-art uncertainty estimation (Table 1: average AUSE of 0.18 on depth, NLL of -0.74 on RGB for LF dataset)
- Novel view synthesis quality is not sacrificed for uncertainty: PSNR of 30.05 on LF dataset, outperforming baselines by significant margins

**Questions:**

- Only covariance scales are stochastic; opacity, color (SH), and positions appear deterministic. Are there failure modes where scale uncertainty alone is insufficient?

---

> ### Author Response · Authors · 2025-11-12
> **Clarification Regarding Listed Weaknesses**
>
> Dear Reviewer WvZK,
>
> Thank you very much for your thoughtful review and detailed comments. We noticed that the “Weaknesses” section appears to be identical to the “Strengths” section, which may have been a copy-and-paste error. Could you kindly update the weaknesses you intended to highlight? This would greatly help us address your concerns accurately in our rebuttal and revision.
>
> Best,
>
> HorseshoeSplatting Authors

---

> > ### Comment · Reviewer_WvZK · 2025-11-12
> >
> > My apologies for the confusion and thank you for pointing this out. I have now updated my review with the intended weaknesses.

---

> > > ### Author Response · Authors · 2025-11-12
> > >
> > > Thank you very much for the quick updates. We will prepare the rebuttal accordingly

---

> ### Author Response · Authors · 2025-11-23
>
> We sincerely thank the reviewer for the positive assessment, particularly for recognizing our justification of the Horseshoe prior and SOTA performance. We have revised our paper accordingly. Here are our responses to your comments.
>
> >**[W1]** LLFF are small, static, and relatively clean. No tests on large‑scale, outdoor, high‑specular, or OOD settings (e.g., Tanks\&Temples, Mip‑NeRF360)
>
> We sincerely thank the reviewer for this valuable suggestion. We agree that validating on diverse and challenging scenarios is essential.
> To address this, we have conducted comprehensive additional experiments on the Tanks&Temples and Mip-NeRF360 datasets, which feature large-scale, unbounded outdoor scenes with complex geometry and varying lighting.
>
> The results are presented in **Table R1**, and it is observed that our method demonstrates strong generalization capabilities.
> First, our method outperforms baselines across all datasets in uncertainty estimation. Notably, on Tanks&Temples, our method reduces the NLL from 2.46 to 0.58, and on Mip-NeRF360 from 2.88 to 0.72. This confirms that our Horseshoe prior can effectively handle structural sparsity even in complex **wild** settings.
> Moreover, we maintain or exceed the rendering quality (PSNR/SSIM/LPIPS) of competing uncertainty-aware methods.
>
> We have included these results in the revised manuscript to demonstrate the broad applicability of our approach.
>
> **Table R1: Novel View Synthesis (NVS) and Uncertainty Estimation on Tanks & Temples and Mip-NeRF360. Our method achieves significantly better uncertainty metrics (AUSE, NLL) while maintaining superior visual quality.**
> | Dataset | Method | PSNR ↑ | SSIM ↑ | LPIPS ↓ | AUSE ↓ | NLL ↓ |
> | :--- | :--- | :---: | :---: | :---: | :---: | :---: |
> | **Tanks&Temples** | FisherRF | 23.41 | 0.837 | 0.198 | 0.57 | 5.94 |
> | | Variational 3DGS | 23.45 | 0.835 | 0.199 | 0.47 | 2.46 |
> | | **Ours** | **23.67** | **0.845** | **0.186** | **0.35** | **0.58** |
> | **Mip-NeRF360** | FisherRF | 27.36 | 0.803 | 0.238 | 0.56 | 6.11 |
> | | Variational 3DGS | 27.28 | 0.797 | 0.250 | 0.51 | 2.88 |
> | | **Ours** | **27.68** | **0.810** | **0.207** | **0.35** | **0.72** |
>
> >**[W2]** The paper does not give a clear motivation for use 3DGS to model uncertainty in the storyline. So it's hard for me to understand why radiance field rendering needs uncertainty quantification. Also authors should provide comparison with SOTA non-racence field based uncertainty methods. Please include more discussion in introduction and related work section.
>
> We thank the reviewer for this valuable suggestion to strengthen the narrative and context of our work.
> Following your advice, we have enhanced the motivation and positioning of our work in both the **Introduction** and **Related Work** sections.
>
> We have revised the **Introduction** section to (i) **articulate that calibrated uncertainty improves robustness under sparse views and occlusions and enables active view selection/mapping**, and (ii) **position our hierarchical prior as a principled Bayesian approach in explicit 3DGS.**
> Prior work in NeRF shows that uncertainty reduces artifacts and guides data acquisition, and recent works explicitly use UQ (Uncertainty Quantification) for active mapping; we bring these benefits to 3DGS with a structural, shrinkage-based prior on covariance scales that targets the source of rendering ambiguity.
>
> We have also expanded our **Related Work** section to explicitly discuss and compare with SOTA non-radiance field uncertainty methods in which confidence is standard in MVS depth fusion, clarifying the broader relevance.

---

> > ### Author Response · Authors · 2025-11-23
> >
> > >**[Q1]** Only covariance scales are stochastic; opacity, color (SH), and positions appear deterministic. Are there failure modes where scale uncertainty alone is insufficient?
> >
> > Thanks for the important question. In 3DGS, per-pixel color is a weighted blend of nearby splats whose weights are dominated by each splat’s screen-space footprint—i.e., the projection of its 3D covariance—and the ensuing $\alpha$-compositing order. Consequently, the primary drivers of uncertainty under sparse views or depth ambiguity are the **axis-wise scales** that set those footprints; small changes in scales modulate coverage, overlap, and visibility contributions more than any other per-splat parameter. This is exactly what our stochastic scales target: by sampling the scales (with fixed $(\mu, R)$) we capture the dominant uncertainty in who contributes to each pixel and by how much, while keeping the renderer and rasterization faithful to 3DGS’ formulation. Prior work corroborates that Gaussians’ shape (covariance) governs tile culling, sorting, and blending, and that many artifacts and pruning decisions arise from footprint/opacity interplay during rasterization and densification, not from color coefficients alone. Our experiments show that scale-only stochasticity already yields calibrated, useful uncertainty maps without degrading real-time performance; modeling color/opacity is a natural extension but not required to address the structural sparsity and coverage uncertainties we study.
> >
> > While scale-based uncertainty effectively resolves geometric and coverage ambiguity, specific scenarios dominated by purely photometric variance—such as extreme specularities, color calibration drifts, or complex semi-transparent structures—present instances where uncertainty stems from SH coefficients or opacity rather than footprint coverage. For such cases, our method can be naturally extended by placing (i) band-wise global–local shrinkage priors on SH coefficients to stabilize view-dependent appearance, and (ii) structured priors (e.g., Logit-Normal) on opacity to refine the modeling of thin structures. This highlights the versatility of our formulation to adapt to highly complex photometric phenomena in future work.

---

### Author Response · Authors · 2025-12-03
**Summary of Discussion**

We sincerely thank the reviewers for their thoughtful feedback and constructive suggestions. We are encouraged by the overall positive reception and by the acknowledgement that our method addresses a key limitation in existing 3DGS pipelines.

We appreciate the reviewers’ recognition of our contributions, who noted that our work provides **a principled Bayesian treatment** of structural sparsity and uncertainty in 3DGS **(Reviewers WvZK, xYVt, QC4h)**, supported by **theoretical guarantees** **(Reviewers WvZK, QC4h, W9qb)** and **strong empirical performance with modest computational overhead** **(Reviewers WvZK, xYVt, QC4h, W9qb)**.

**Note on Reviewer WvZK’s comments.**
In the initial review, the *Weaknesses* section inadvertently duplicated the *Strengths*. After we sought clarification, the reviewer updated the review with questions on dataset scale and motivation. Due to the recent data leak incident, the visible review may have reverted; our rebuttal and the summary below respond to the corrected version.

Below we summarize how we address the main concerns and the resulting improvements made based on the reviewers’ suggestions:

1. **How we address main concerns on scope, scalability, and robustness.**
   To clarify the **conceptual scope and contribution**, we explicitly framed our work as addressing a *central modeling deficiency* of 3DGS—structural sparsity and posterior-based uncertainty in covariance scales—via a global–local Horseshoe prior and contraction analysis, and contrasted this with prior pruning-based or offset-based approaches.
   To address concerns about **computational scalability and accessibility**, we conducted a detailed efficiency benchmark on a single RTX 3090, showed that computational cost scales with the number of Gaussians (and thus decreases on smaller datasets), highlighted that our method is **significantly more efficient than ensembles** while preserving real-time rendering, and committed to releasing code and pretrained models to support reproducibility.
   To strengthen **robustness across diverse scenarios**, we extended our experiments to large-scale outdoor datasets (Tanks&Temples, Mip-NeRF360), designed an active view selection setup under severe view sparsity, and introduced an OOD view detection task to evaluate behavior under challenging and out-of-distribution conditions.
   To ensure that the gains are **specifically attributable to the Horseshoe hierarchy**, we performed a prior sweep and sparsity analysis (Gaussian vs. Laplace vs. Horseshoe), demonstrating that the Horseshoe prior is *not merely any sparsity prior* but induces stronger structural sparsity and better-calibrated uncertainty while preserving reconstruction quality.

2. **Theoretical and modeling refinements.**
   We strengthened the motivation for uncertainty quantification in 3DGS by linking it to robustness under sparse views, occlusions, and active view selection/mapping, and by connecting our formulation to prior NeRF-based UQ literature. We made the assumptions in our contraction analysis explicit by stating the local depth- and activity-margin conditions under which the renderer is locally Lipschitz. We clarified why the hierarchical Horseshoe prior is placed on covariance scales—parameters that control screen-space footprints and visibility—and provided quantitative evidence of induced structural sparsity via a prior sweep over Gaussian/Laplace/Horseshoe and sparsity statistics on learned scales (**Table 5, Table 7, Figure 4 in the revised manuscript**), showing that the Horseshoe is not “just any” sparsity prior but yields stronger structural sparsity and better-calibrated uncertainty.

3. **Expanded empirical evaluation and practical utility of uncertainty.**
   We broadened the empirical evaluation beyond small, clean benchmarks by adding large-scale, unbounded outdoor datasets (Tanks&Temples, Mip-NeRF360), where our method consistently improves uncertainty calibration (AUSE, NLL) while matching or slightly improving PSNR/SSIM/LPIPS over uncertainty-aware baselines (**Table 8**). To demonstrate practical utility, we detailed and extended the active view selection protocol (acquisition function, shared budget, stopping criterion) from LLFF to Tanks&Temples, where uncertainty-guided selection achieves better reconstruction quality than random and variational baselines (**Table 10**), and we introduced an OOD view detection task on LLFF, where using uncertainty as an anomaly score yields substantially higher AUROC than prior methods (**Table 11**). We also reported training time, memory usage, FPS, and model size on a single RTX 3090 (**Table 9**), showing that our factorized variational scheme adds only modest training overhead while preserving real-time rendering and remaining far more efficient than ensemble-based alternatives.

Thank you again for your time and feedback, which helped us substantially strengthen the clarity, scope, and empirical support of the paper.

---

### Meta-Review · Area_Chair_2KY1 · 2025-12-31

**Summary:**

# Decision

This submission presents a well-founded exploration of uncertainty modeling in 3DGS, introducing Bayesian shrinkage and structured sparsity to address noise-dominated covariance components. While reviewers initially showed concerns regarding the scope and practical impact, they acknowledged its strong theoretical grounding and principled probabilistic formulation.

Importantly, the authors substantially strengthened the paper during rebuttal. They provided additional evaluations that directly addressed major concerns, including a clearer analysis of the role of structural sparsity, new results on more recent benchmarks (e.g., Mip-NeRF-360 and Tanks&Temples), an evaluation of the computational overhead introduced by the variational inference scheme, etc. These additions significantly improve the clarity, credibility, and empirical support of the proposed approach.

While limitations remain (e.g., focus on covariance parameters), the combination of solid theory and improved experimental validation raises the contribution above the acceptance threshold. Overall, the paper represents a meaningful and rigorous step toward uncertainty modeling in 3DGS. We recommend acceptance.

------------
# Consolidated Reviews

## Strengths

### Relevant and solidly-grounded contributions
- Valuable exploration of uncertainty modeling in 3DGS [`WvZK`, `xYVt`, `QC4h`, `W9qb`].
- Technical novelty: incorporating both uncertainty and structured sparsity, c.f. insufficient regularization of noise-dominated components in covariance structures [`WvZK`, `xYVt`, `QC4h`].
- Solid theoretical grounding [`QC4h`, `W9qb`].

### Decent results on provided evaluation
- SOTA results on uncertainty estimation [`WvZK`, `xYVt`, `QC4h`].
- Good results on NVS evaluation [`WvZK`, `W9qb`].
- Decent ablation (e.g., Gaussian vs. Horseshoe) [`xYVt`].


## Weaknesses

### Scope of the contributions remained unclear to some reviewers
- Solution applied only to covariance scales and not other 3DGS parameters [`WvZK`, `W9qb`].
- Limited conceptual novelty beyond integration of known Bayesian shrinkage techniques [`QC4h`].
- Unclear computational overhead brought by the variational inference [`QC4h`].
- More clarification or evidence needed w.r.t. local Lipschitz assumption and handling of discrete transitions [`xYVt`].
- Poor justification for uncertainty quantification in 3DGS [`WvZK`].

### Limited real-world evaluation
- Lack of evaluation on more challenging/real-world benchmarks [`WvZK`, `xYVt`, `QC4h`].
- Lack of quantitative evaluation of the benefits of uncertainty quantification to downstream tasks (besides overly-simple active-view selection experiment) [`QC4h`, `W9qb`].
- Lack of details w.r.t. active-view selection strategy [`xYVt`].
- Limited ablation w.r.t. benefits of Horseshoe vs. sparsity-prior introduction [`W9qb`].

**Reviewer Concerns:**

See above for summary of main concerns shared by reviewers.

During rebuttal, the authors addressed several of the key concerns raised by the reviewers, especially by significantly expanding the experimental validation.
They provided a more explicit evaluation of structural sparsity, added results on more challenging / widely-used benchmarks (Mip-NeRF-360 and Tanks&Temples), quantified the computational overhead of the proposed variational inference framework, and provided additional downstream task experiments.

**Reviewer Scores:**

### Reviewer `WvZK`
- **Original score:** 6
- **Score change:** likely to have kept their score, c.f. low confidence and superficial understanding/review of the paper.

### Reviewer `xYVt`
- **Original score:** 6
- **Score change:** (similarly) likely to have kept their score, c.f. low confidence and superficial understanding/review of the paper.

### Reviewer `QC4h`
- **Original score:** 4
- **Score change:** might have increased to ~6, c.f. concerns mostly covered by the authors' response ; though likely to have remained unresponsive.

### Reviewer `W9qb`
- **Original score:** 6
- **Score change:** might have kept their score or slightly increased, c.f. reviewer's positive response to the authors' rebuttal.

---

### Decision · Program_Chairs · 2026-01-26

Accept (Poster)